# MDGMIX: Boundary-Aware Subgraph Mixing for Multi-Domain Graph Pre-Training

Ziyu Zheng [1]  Yaming Yang [1]  Ziyu Guan [1]  Wei Zhao [1]  Xinyan Huang [2]

## Abstract

Multi-domain graph pre-training is a crucial step in constructing foundational graph models with cross-domain generalization capabilities. However, existing methods predominantly rely on jointly training all source domain graphs, resulting in high computational costs. Furthermore, it remains unclear whether all source domain graph data contribute equally to effective transfer. This paper empirically reveals significant data redundancy in multi-domain graph pre-training. Based on this finding, we propose the Multi-domain Graph Pre-training Framework, MDGMIX, which combines boundary-aware subgraph mixing with hierarchical discrimination. By selecting boundary nodes to construct challenging mixed-domain subgraphs, MDGMIX employs coarse-grained domain discrimination and fine-grained domain decomposition losses to decouple shared patterns from domain-specific patterns. During adaptation, MDGMIX employs a lightweight prompt weighting mechanism to transfer source domain knowledge. Extensive experiments demonstrate that MDGMIX consistently outperforms strong baselines in few-shot classification tasks while exhibiting superior time and memory efficiency. The code is available at: https://github.com/zhengziyu77/MDGMIX.

## 1. Introduction

Graph-structured data naturally captures complex relationships between entities and is widely applied in fields such as social networks (Fan et al., 2019), recommendation systems (Wu et al., 2022), and molecular design (Reiser et al.,

[1] School of Computer Science and Technology, Xidian University, Xi'an, China [2] School of Artificial Intelligence, Xidian University, Xi'an, China. Correspondence to: Wei Zhao <ywzhao@mail.xidian.edu.cn>.

*Proceedings of the 43rd International Conference on Machine Learning*, Seoul, South Korea. PMLR 306, 2026. Copyright 2026 by the author(s).

2022). In recent years, inspired by the success of foundation models in natural language processing (Paaß & Giesselbach, 2023) and computer vision (Awais et al., 2025), the field of graph learning has begun exploring graph foundation models (Liu et al., 2025; Ding et al., 2023; Liu et al., 2022). These models aim to learn transferable, general knowledge from large-scale, multi-domain graph data through pre-training, thereby advancing toward a more universal framework for graph learning.

Although existing research has begun to explore multi-domain graph pre-training (Zhao et al., 2024; Yuan et al., 2025; Wang et al., 2025), most current methods still follow pre-training strategies designed for single domains (Yu et al., 2024; 2025; Sun et al., 2022), overlooking the substantial computational overhead incurred by jointly training on all data. This leads to serious scalability and efficiency challenges. As illustrated in Figure 1, as the number of source domains increases, joint pre-training based on multi-domain graphs often results in a sharp rise in memory or time consumption, severely limiting its practicality. This points to a deeper underlying issue: Due to the distributional shifts between source and target domains, the actual benefit of multi-domain graph pre-training data merits careful scrutiny—is all pre-training graph data equally conducive to cross-domain generalization?

To investigate this question, we randomly drop varying proportions of nodes from the multi-domain graph data and observe its impact on the target domain performance. Figure 2 reveals a critical and counter-intuitive phenomenon: even after randomly discarding up to 90% of the pre-training graph data, the model only exhibits marginal performance degradation across multiple downstream benchmarks. This result indicates severe data redundancy in multi-domain graph pretraining settings: the existing paradigm consumes substantial computational resources on graph data that contributes little to transfer learning, while failing to focus on knowledge relevant for meaningful cross-domain generalization adequately.

These observations suggest that achieving efficient and reliable multi-domain graph pre-training does not depend solely on increasing the scale of pre-training data, but rather on constructing pre-training data and objectives with higher trans-

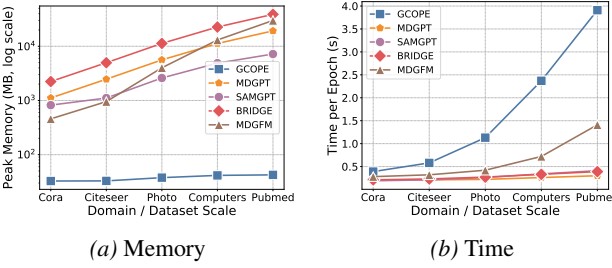

*Figure 1.* Efficiency analysis of multi-domain graph methods.

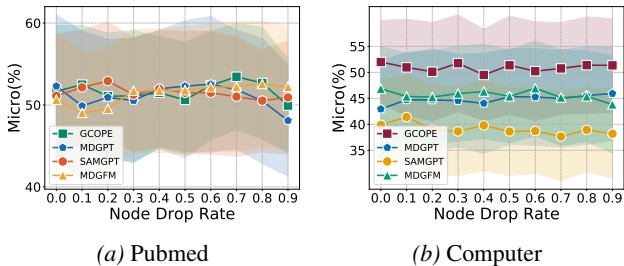

*Figure 2.* Empirical analysis of graph data redundancy.

fer utility. The core objective of multi-domain graph pre-training is to learn domain-invariant representations across multiple source domains, enabling stable performance on unseen target domains. Inspired by classical domain generalization studies (Muandet et al., 2013; Bui et al., 2021), transferable knowledge is often characterized by invariant features shared across domains (Li et al., 2022), which typically concentrate in regions where domain discrimination is most challenging and uncertain—specifically, near the domain boundaries (Wang et al., 2022).

Thus, we propose MDGMIX, a multi-domain graph pre-training framework based on subgraph mixing and hierarchical discrimination. On the data side, MDGMIX identifies structurally similar boundary nodes prone to cross-domain confusion and performs subgraph mixing around them to construct challenging training samples. This strategy enables the model to capture shared and domain-specific structures without redundantly modeling full source graphs. On the objective side, we introduce a hierarchical discrimination objective that combines coarse-grained domain discrimination with fine-grained domain decomposition, facilitating the separation of domain-invariant knowledge from domain-specific patterns within mixed subgraphs. For cross-domain adaptation, instead of introducing domain-specific tokens during pre-training, we adopt a lightweight prompt-weighting mechanism that integrates source-domain centroids into target representations via learnable weights, requiring only a small number of trainable parameters and improving training and inference efficiency.

In summary, our contributions are as follows:

- We reveal substantial data redundancy in multi-domain graph pre-training, showing that large portions of jointly pre-trained source-domain graphs are empirically redundant for cross-domain transfer.

- We propose MDGMIX, a multi-domain graph pre-training framework that selects boundary nodes and performs cross-domain subgraph mixing, together with hierarchical domain discrimination.

- Extensive experiments show that MDGMIX achieves

superior few-shot cross-domain performance with improved time and memory efficiency.

## 2. Related Work

Multi-domain graph foundation models (Liu et al., 2025; 2024; Yuan et al., 2026) seek to learn generalizable structural and semantic knowledge from multiple source domains for effective transfer to unseen target domains. Existing methods follow a two-stage paradigm of pre-training and adaptation. For instance, GCOPE (Zhao et al., 2024) introduces virtual nodes to bridge structural gaps, MDGPT (Yu et al., 2024) assigns domain tokens to enhance discriminability, SAMGPT (Yu et al., 2025), SA2GFM (Shi et al., 2025), and MDGFM (Wang et al., 2025) model structural differences with domain-specific priors, and BRIDGE (Yuan et al.) employs alignment risk regularization and MoE-based experts to strengthen transfer. However, these approaches uniformly rely on full-source-domain pre-training, ignoring inherent data redundancy and incurring high computational cost while necessitating complex adaptation modules to mitigate cross-domain discrepancies. For more related work, see the Appendix B.

## 3. Preliminary

**Notation.** Given $K$ graph domains $\{\mathcal{G}^{(k)}\}_{k=1}^{K}$, where the $k$-th domain is denoted as $\mathcal{G}^{(k)} = (\mathcal{V}^{(k)}, \mathcal{E}^{(k)}, \mathbf{X}_{\text{init}}^{(k)})$, with $\mathcal{V}^{(k)}$ being the node set, $\mathcal{E}^{(k)}$ the edge set, and $\mathbf{X}_{int}^{(k)} \in \mathbb{R}^{|\mathcal{V}^{(k)}| \times d^{(k)}}$ the node feature matrix. Each graph is associated with a domain. In the downstream phase, given a target graph $\mathcal{G}^{(t)}$, the parameters of the GNN are frozen, and the generalization ability of the pre-trained model is tested on unseen graphs.

**Mixup.** Mixup (Zhang et al., 2017) is a data augmentation strategy originally proposed for image classification, which constructs virtual training samples by linearly interpolating pairs of examples in both feature and label spaces. Given two labeled samples $(\mathbf{x}_i, \mathbf{y}_i)$ and $(\mathbf{x}_j, \mathbf{y}_j)$, Mixup generates

a new sample $(\tilde{\mathbf{x}}, \tilde{\mathbf{y}})$ as

$$\tilde{\mathbf{x}} = \lambda \mathbf{x}_i + (1 - \lambda)\mathbf{x}_j, \tag{1}$$

$$\tilde{\mathbf{y}} = \lambda \mathbf{y}_i + (1 - \lambda)\mathbf{y}_j, \tag{2}$$

where $\lambda \in [0, 1]$ is a mixing coefficient typically drawn from a Beta distribution $\text{Beta}(\alpha, \alpha)$ with $\alpha > 0$. This symmetric distribution controls the interpolation strength: smaller $\alpha$ concentrates $\lambda$ near 0 or 1, yielding samples close to the original data, while larger $\alpha$ encourages stronger mixing.

## 4. Method

In this work, we present MDGMIX, a subgraph-level multi-domain graph pre-training framework that improves efficiency by pre-training on domain-ambiguous regions rather than modeling all source-domain graphs, as shown in Figure 3. Formally, the framework consists of four components: (1) multi-domain boundary node selection, (2) multi-domain subgraph mixing, (3) multi-level domain discrimination pre-training, and (4) cross-domain adaptation.

### 4.1. Boundary Node Selection

Since the raw feature dimensions typically differ across domains, we first project all node features into a unified embedding space following prior work (Yu et al., 2025):

$$\mathbf{X}^{(k)} = \text{Proj}(\mathbf{X}_{\text{init}}^{(k)}), \tag{3}$$

where $\mathbf{X}^{(k)} \in \mathbb{R}^{|V^{(k)}| \times d}$ denotes the aligned feature matrix, and $\text{Proj}(\cdot)$ corresponds to PCA-based dimensionality reduction (Abdi & Williams, 2010).

We then compute a domain center for each graph domain. For domain $k$, the center vector $\mathbf{c}_k$ is defined as the mean feature of all nodes within the domain:

$$\mathbf{c}_k = \frac{1}{|\mathcal{V}^{(k)}|} \sum_{v_i \in \mathcal{V}^{(k)}} \mathbf{x}_i^{(k)}, \tag{4}$$

where $\mathbf{x}_i^{(k)} \in \mathbb{R}^d$ denotes the feature vector of domain node $v_i$. The domain center serves as a reference for measuring node-to-domain proximity.

To identify nodes that are easily confused across domains, we propose a multi-domain consensus boundary detection mechanism that quantifies the relative margin between a node and multiple domain centers. For a node $v_i \in \mathcal{G}^{(k)}$, its distance to the center of domain $k$ is defined as:

$$d_{i,k} = \|\mathbf{x}_i - \mathbf{c}_k\|_2. \tag{5}$$

For each domain pair $(k, m)$ with $k \neq m$, we compute a pairwise margin:

$$\Delta_i^{(k,m)} = |d_{i,k} - d_{i,m}|, \tag{6}$$

where smaller absolute margins indicate higher boundary ambiguity between domains $\mathcal{G}^{(k)}$ and $\mathcal{G}^{(m)}$. Based on this margin, We compute confidence scores after min-max normalizing the pairwise margin.

$$s_i^{(k,m)} = 1 - \text{Min-Max}(\Delta_i^{(k,m)}). \tag{7}$$

For each domain pair $(k, m)$, we select the top $\lceil \rho \cdot |\mathcal{V}^{(k)}| \rceil$ nodes with the highest confidence as the candidate boundary set $\mathcal{B}_{k,m}$, where $\rho \in (0, 1)$ is a hyperparameter controlling the proportion of boundary nodes.

To ensure that boundary nodes exhibit ambiguity across multiple domains pairs, we take the intersection of all pairwise candidate sets:

$$\mathcal{B}_k = \bigcap_{m \neq k} \mathcal{B}_{k,m}. \tag{8}$$

If the intersection is empty, we instead adopt the union of all candidate sets as a fallback strategy. By focusing on nodes that are simultaneously ambiguous across multiple domain pairs, MDGMIX explicitly concentrates pre-training on regions that are most informative for cross-domain transfer.

### 4.2. Domain-Mixup Subgraph Construction

Given boundary nodes from different domains, we construct mixed-domain subgraphs to generate challenging training samples that interpolate between domain distributions. For two boundary nodes $v_i \in \mathcal{G}^{(k)}$ and $v_j \in \mathcal{G}^{(m)}$ selected from the boundary sets $\mathcal{B}$, we compute their cosine similarity:

$$\text{sim}(\mathbf{x}_i^{(k)}, \mathbf{x}_j^{(m)}) = \frac{\mathbf{x}_i^{(k)\top} \mathbf{x}_j^{(m)}}{\|\mathbf{x}_i^{(k)}\| \cdot \|\mathbf{x}_j^{(m)}\|}. \tag{9}$$

Given a similarity threshold $\gamma$, we select $N$ high-similarity cross-domain node pairs from the boundary nodes satisfying $\text{sim}(\mathbf{x}_i, \mathbf{x}_j) > \gamma$ to construct mixed-domain samples.

For each selected pair, we extract a $k$-hop subgraph centered at the sampled node, forming $\mathcal{G}^{\text{sub}} = (\mathcal{V}^{\text{sub}}, \mathcal{E}^{\text{sub}}, \mathbf{X}^{\text{sub}})$. Inspired by mixup (Zhang et al., 2017; Lu et al., 2025), we adopt a center-based subgraph mixup strategy to construct inter-domain mixed subgraphs.

Given two $k$-hop subgraphs $\mathcal{G}_{i,k}^{\text{sub}}$ and $\mathcal{G}_{j,m}^{\text{sub}}$ from different domains, the mixed inter-domain subgraph is defined as

$$\mathcal{G}_{ij}^{\text{inter}} = \mathcal{M}_g(\mathcal{G}_{i,k}^{\text{sub}}, \mathcal{G}_{j,m}^{\text{sub}}, \lambda) = (\mathcal{V}_{ij}^{\text{inter}}, \mathcal{E}_{ij}^{\text{inter}}, \tilde{\mathbf{X}}_{ij}^{\text{inter}}). \tag{10}$$

For the center nodes of the two subgraphs, we apply a mixup operation to obtain the mixed center feature:

$$\tilde{x}_{ij} = \lambda \, x_{i,k}^{\text{sub}} + (1 - \lambda) \, x_{j,m}^{\text{sub}}, \tag{11}$$

where $x_{i,k}^{\text{sub}}$ and $x_{j,m}^{\text{sub}}$ denote center-node features from domains $D_k$ and $D_m$, respectively. For the mixed graph structure, we perform a union operation on the neighborhoods of

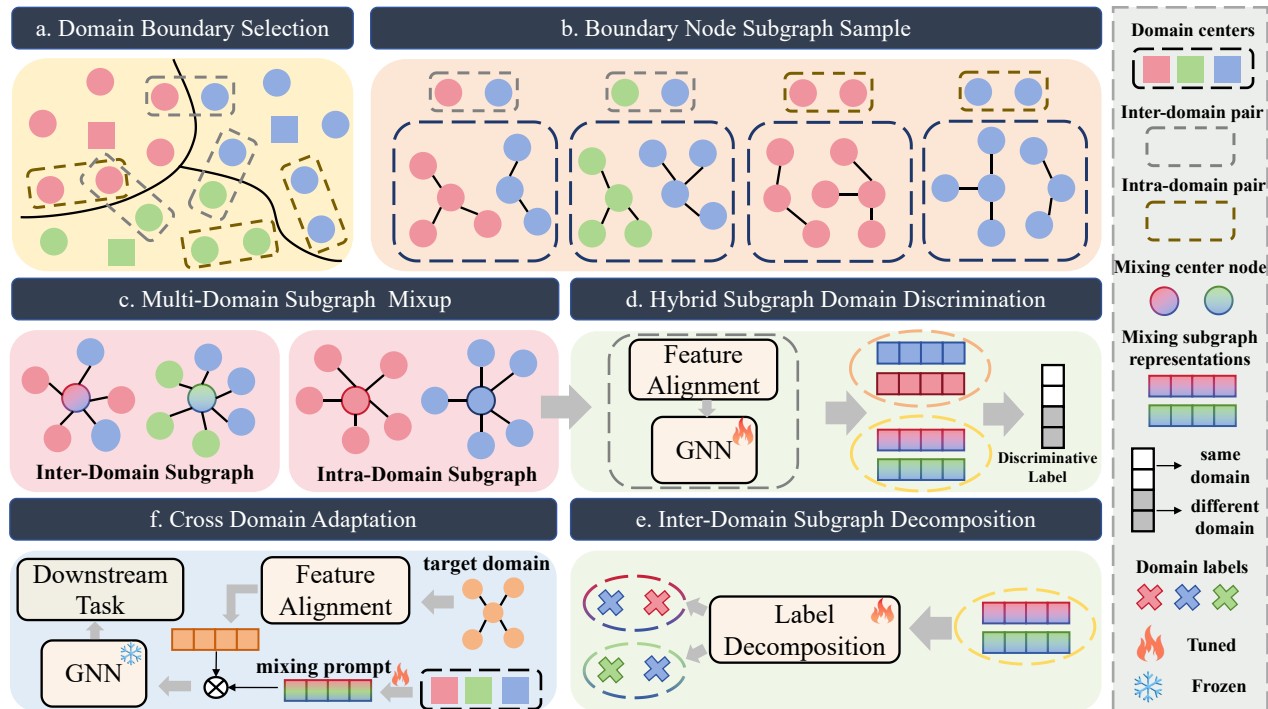

Figure 3. The overall framework of the MDGMIX.

different subgraphs to preserve the structural information of the original subgraphs.

We use a fixed $\lambda = 0.5$ to ensure symmetric contributions from both domains, which is particularly important since selected boundary nodes are already highly ambiguous across domains. We construct mixed-domain labels for pretraining:

$$\tilde{y}_{ij} = \lambda\, y_{i,k}^{\text{sub}} + (1 - \lambda)\, y_{j,m}^{\text{sub}}, \qquad (12)$$

where $\tilde{y}_{ij} \in \mathbb{R}^K$ is a multi-class label whose components correspond to the two mixed domains. These labels are used in the inter-domain subgraph decomposition loss.

To prevent loss of domain-specific information, we also construct intra-domain mixed subgraphs. For each domain $D_k$, we randomly sample $N/K$ same-domain node pairs and construct intra-domain mixed subgraphs:

$$\mathcal{G}_{ij}^{\text{intra}} = \mathcal{M}_g(\mathcal{G}_{i,k}^{\text{sub}}, \mathcal{G}_{j,k}^{\text{sub}}, \lambda) = (\mathcal{V}_{ij}^{\text{intra}}, \mathcal{E}_{ij}^{\text{intra}}, \tilde{\mathbf{X}}_{ij}^{\text{intra}}), \quad (13)$$

where the mixing operation is identical to the inter-domain case. Different from prior methods (Liu et al., 2025) that depend on full datasets from each domain, our approach leverages interpolations between cross-domain samples to construct lightweight mixed-domain subgraphs, thereby enhancing the efficiency of multi-domain graph pre-training.

### 4.3. Multi-level Domain Discrimination Pretraining

**Hybrid Subgraph Domain Discrimination.** Following the construction described above, the original multi-domain

dataset is compressed into a mixed-domain subgraph dataset $\mathcal{G}^{\text{mix}}(\mathcal{G}^{\text{inter}}, \mathcal{G}^{\text{intra}})$, consisting of $N$ inter-domain mixed subgraphs and $N$ intra-domain mixed subgraphs. To capture coarse-grained domain differences, we assign label 0 to intra-domain subgraphs and label 1 to inter-domain subgraphs, resulting in binary coarse-grained domain labels $\mathbf{Y}^{\text{mix}} \in \{0, 1\}^{2N}$. For each mixed subgraph, its representation is computed as:

$$\tilde{\mathbf{h}}_{ij} = \mathcal{P}\big(\text{GNN}(\mathcal{G}_{ij}^{\text{mix}})\big), \qquad (14)$$

where $\tilde{\mathbf{h}}_{ij}$ denotes the graph-level representation of the mixed subgraph composed of subgraphs $i$ and $j$, and $\mathcal{P}$ represents a graph pooling operation. A domain discriminator is then used to determine whether a mixed subgraph belongs to the inter-domain class:

$$\hat{\mathbf{y}}_{ij} = D_\psi(\tilde{\mathbf{h}}_{ij}), \qquad (15)$$

where $D_\psi$ is a linear classifier (with softmax) predicting the coarse-grained domain class, and $\hat{y}_{ij}$ denote the predicted probabilities. The domain discrimination loss is defined as

$$\mathcal{L}_{\text{dis}} = -\mathbb{E}_{(i,j)}\Big[ y_{ij}^{\text{mix}} \log(\hat{y}_{ij}) + (1 - y_{ij}^{\text{mix}}) \log(\hat{y}_{ij})\Big]. \quad (16)$$

We learn coarse-grained domain boundaries by classifying inter-domain subgraphs and intra-domain subgraphs.

**Inter-Domain Subgraph Decomposition.** To further distill fine-grained domain-shared knowledge across domains

and suppress explicit domain-specific bias in subgraph representations, we introduce an adversarial cross-domain subgraph label decomposition strategy.

Specifically, based on the mixed-domain subgraph predictions obtained from the coarse-grained discriminator, we introduce a binary gating mechanism to select inter-domain mixed subgraphs for fine-grained domain decomposition.

Let $\hat{y}_{ij}^{\mathrm{mix}} \in \mathbb{R}^2$ denote the predicted probability vector indicating whether a mixed subgraph $G_{ij}$ is intra-domain or inter-domain. We define the subgraph encoding mask as

$$\mathbf{m}_{ij} = \mathbb{1}\big[\hat{y}_{ij,\,\mathrm{inter}}^{\mathrm{mix}} > 0.5\big], \tag{17}$$

where $\mathbf{m}_{ij} \in \{0, 1\}$ is a binary indicator that activates only for inter-domain mixed subgraphs.

Since the mixed-domain labels $\tilde{\mathbf{Y}}_{ij}$ defined in Eq. (12) represent a continuous domain composition distribution rather than discrete categorical labels, we adopt the KL divergence to model distribution-level consistency. For an inter-domain mixed subgraph, the predicted domain proportion distribution is computed as

$$\mathbf{p}_{ij} = Q_\phi(\mathrm{GRL}(\tilde{\mathbf{h}}_{ij} \cdot \mathbf{m}_{ij})), \tag{18}$$

where $Q_\phi$ denotes the domain label decomposition network parameterized by $\phi$, and $\mathrm{GRL}(\cdot)$ represents a Gradient Reversal Layer. The gradient reversal layer enforces the encoder to suppress explicit domain-identifiable signals while preserving the decomposer's ability to recover domain proportions. The binary mask $\mathbf{m}_{ij}$ is applied at the loss level to ensure that the fine-grained decomposition loss is computed only for inter-domain mixed subgraphs. The fine-grained distribution prediction loss is defined as:

$$\mathcal{L}_{\mathrm{fine}} = \frac{1}{|\mathcal{M}|} \sum_{(i,j)\in\mathcal{M}} \sum_{d=1}^{K} \tilde{y}_{ij}^{(d)} \log \frac{\tilde{y}_{ij}^{(d)}}{p_{ij}^{(d)}}, \tag{19}$$

where $\mathcal{M}$ denotes the set of predicted inter-domain mixed subgraphs, $\tilde{y}_{ij}^{(d)}$ is the ground-truth domain proportion, and $p_{ij}^{(d)}$ is the predicted proportion. Under this mechanism, the domain decomposer is encouraged to accurately recover the domain composition distribution of each mixed subgraph, while the shared encoder is forced to produce subgraph representations that are indistinguishable with respect to their domain origins, thus facilitating the learning of domain-invariant representations across domains.

**Multi-level Domain Discrimination Loss**  Combining the coarse-grained and fine-grained objectives, the overall pre-training loss is

$$\mathcal{L}_{\mathrm{pre}} = \mathcal{L}_{\mathrm{dis}} + \mathcal{L}_{\mathrm{fine}}. \tag{20}$$

The coarse-grained loss $\mathcal{L}_{\mathrm{dis}}$ learns the basic patterns distinguishing intra- and inter-domain mixtures, while the fine-grained decomposition loss $\mathcal{L}_{\mathrm{fine}}$ explicitly models the domain composition of each mixed sample. This multi-level supervision mechanism enables the model to capture cross-domain relationships at different granularities, providing richer domain-invariant feature representations for subsequent domain generalization tasks.

### 4.4. Cross Domain Adaptation

During adaptation, we freeze the pretrained GNN and transfer source-domain knowledge through a lightweight prompt-weighting mechanism. Instead of introducing domain-specific tokens during pre-training, we leverage source-domain centers $\{c_i\}_{i=1}^{K}$ and compute a mixed prompt:

$$\mathbf{p}^{\mathrm{mix}} = \sum_{i=1}^{K} \alpha_i \mathbf{c}_i, \tag{21}$$

where only the weighting coefficients $\alpha_i$ are learnable. Then, we use the mixed prompt to enhance node features of the target domain. The augmented node is represented as:

$$H = \mathrm{GNN}(\mathcal{G}^t, \mathbf{X}^t \odot \mathbf{p}^{\mathrm{mix}}), \tag{22}$$

where $\mathbf{X}^t$ is the aligned target domain feature. For downstream node or graph classification tasks, we follow prior designs and adopt a generic task template based on subgraph similarity. The loss function is defined as:

$$\mathcal{L}_{\mathrm{ds}} = - \sum_{(x_i, y_i)} \ln \frac{\exp\left(\mathrm{sim}(\mathbf{h}_{x_i}, \hat{\mathbf{h}}_{y_i})/\tau\right)}{\sum_{c\in\mathcal{Y}} \exp\left(\mathrm{sim}(\mathbf{h}_{x_i}, \hat{\mathbf{h}}_c)/\tau\right)}, \tag{23}$$

where $\tau$ is the temperature parameter and $\mathrm{sim}(\cdot)$ denotes cosine similarity. $\mathbf{h}_{x_i}$ represents the final node or graph embedding, and $\hat{\mathbf{h}}_c$ denotes the class prototype of class $c$, computed as the mean representation of labeled nodes or graphs of that class. Notably, the number of trainable parameters scales linearly with the number of source domains, making the adaptation phase both efficient and stable.

## 5. Theoretical Analysis

In this section, we provide a theoretical justification for MDGMIX from a multi-source domain generalization perspective (Sicilia et al., 2023; Albuquerque et al., 2019). Our analysis shows that training exclusively on boundary-centered mixed subgraphs suffices for effective cross-domain transfer, with detailed proofs in Appendix C.

### 5.1. Generalization Bound

We analyze the subgraph-level generalization error. Let $\mathcal{H} = \{h = g \circ f_\theta\}$ be the hypothesis class comprising a GNN encoder and a task head. Let $P_t$ be the unseen target

domain distribution. MDGMIX constructs a proxy distribution $\tilde{P}$ by mixing subgraphs centered strictly at boundary nodes to approximate an idealized reference distribution $R^*$, which uniformly mixes all source domains and serves as an intermediate bridge between source and target domains.

**Theorem 5.1** (Domain generalization error bound). *Under standard assumptions of encoder stability and Lipschitz continuity of the loss, structural semantic consistency (Assumption C.3), and non-vanishing boundary mass (Assumption C.4), for any hypothesis $h \in \mathcal{H}$, with probability at least $1 - \delta$, the target risk $\epsilon_{P_t}(h)$ is bounded by:*

$$\epsilon_{P_t}(h) \leq \hat{\epsilon}_{\tilde{P}}(h) + \underbrace{L_\ell L_f \cdot W_1(\tilde{P}, R^*)}_{(I)\ Approximation\ Error}$$

$$+ \underbrace{L_\ell L_f \epsilon_{\text{struct}} + \epsilon_{\text{align}}}_{(II)\ Transfer\ Gap} + \underbrace{\sigma_{dep} \sqrt{\frac{\log(2/\delta)}{2n}}}_{(III)\ Sampling\ Variance} \quad (24)$$

*where $\hat{\epsilon}_{\tilde{P}}(h)$ is the empirical training risk on mixed subgraphs, $L_\ell$ and $L_f$ denote the Lipschitz constants of the loss and encoder, respectively. $W_1$ denotes the 1-Wasserstein distance, $\epsilon_{\text{struct}}$ quantifies structural semantic shift, $\epsilon_{\text{align}}$ captures structural-to-label alignment error, and $\sigma_{dep} = \sqrt{\frac{1}{4} + (n-1)\Delta_{\max}}$ accounts for weak dependence among sampled subgraphs.*

The bound decomposes the target error into three interpretable components, each corresponding to a key design choice in MDGMIX:

1. **Approximation Error** $L_\ell L_f \cdot W_1(\tilde{P}, R^*)$ is controlled by boundary sampling ratio $\rho_{\min}$. Lemma C.8 shows this error scales with $(1 - \rho_{\min})$, theoretically justifying the severe data redundancy observed in Figure 2.

2. **Transfer Gap** $L_\ell L_f \epsilon_{\text{struct}} + \epsilon_{\text{align}}$ captures source-target discrepancy without double-counting structural shift. The decomposition enables generalization under label space shift: $\epsilon_{\text{struct}}$ quantifies domain-invariant structural semantics (Assumption C.3), while $\epsilon_{\text{align}}$ accounts for task-specific label mismatches that require few-shot calibration.

3. **Sampling Variance** $\sigma_{\text{dep}} \sqrt{\log(2/\delta)/2n}$ ensures $\mathcal{O}(1/\sqrt{n})$ convergence under weak dependence (Lemma C.15), despite subgraph overlap from boundary mixing.

**Empirical validation of key assumptions.** We further provide empirical evidence for the key assumptions used in Theorem 5.1. First, to validate the structural–semantic consistency of boundary regions, we train a domain classifier on domain-center subgraphs and evaluate it on center, random,

and boundary subgraphs. Boundary subgraphs lead to the lowest domain classification accuracy and the highest loss, indicating that they are indeed more domain-ambiguous and less dominated by domain-specific signals. Second, we count the number of selected boundary nodes under strict boundary-selection settings and observe non-zero boundary mass across all domain pairs used in our experiments. The detailed results are reported in Appendix C.7.

# 6. Experiment

In this section, we conduct extensive experiments to evaluate the effectiveness of MDGMIX, primarily addressing the following research questions:

- **RQ1.** How does MDGMIX perform under few-shot classification tasks?
- **RQ2.** How do different model components affect overall performance?
- **RQ3.** How efficient is MDGMIX during both pre-training and adaptation compared to existing baselines?
- **RQ4.** How sensitive is the model to hyperparameter variations?

## 6.1. Experimental Setups

**Datasets.** We conduct experiments on eight benchmark datasets spanning diverse domains, including academic networks, e-commerce graphs, and web page networks. Specifically, (1) Academic Networks: Citation networks Cora (Sen et al., 2008), Citeseer (Sen et al., 2008), and Pubmed (Namata et al., 2012), OGBN-Arxiv (Hu et al., 2020).(2) E-commerce Networks: Product co-purchase graphs Photo (Shchur et al., 2018) and Computers (McAuley et al., 2015). (3) Web Page Networks: Hyperlink networks Chameleon and Squirrel (Pei et al., 2020). We also introduce a large-scale graph dataset: the social network Reddit (Hamilton et al., 2017) in Appendix D.3.

**Baselines.** (1) Supervised methods: GCN (Kipf, 2016) and GAT (Veličković et al., 2017). (2) Graph pre-training methods: GraphCL (You et al., 2020) and SimGRACE (Xia et al., 2022). (3) Graph prompting methods: GPF+ (Fang et al., 2023), GraphPrompt (Liu et al., 2023), and Edge-Prompt+ (Fu et al., 2025). (4) Multi-domain graph pre-training methods: GCOPE (Zhao et al., 2024), MDGPT (Yu et al., 2024), SAMGPT (Yu et al., 2025), BRIDGE (Yuan et al.) and MDGFM (Wang et al., 2025).

## Setup of pre-training and downstream tasks.

We evaluate cross-domain generalization to unseen target domains. Each dataset among Cora, Citeseer, Pubmed, Photo, Computers, Chameleon, and Squirrel is treated as

*Table 1.* Performance comparison on node classification tasks (Accuracy). The best results are shown in bold and the runner-ups are underlined.

| Method | Cora | CiteSeer | Pubmed | Photo | Computers | Squirrel | Chameleon | OGBN-Arxiv |
|---|---|---|---|---|---|---|---|---|
| GCN | $37.64_{\pm5.75}$ | $29.61_{\pm5.85}$ | $49.34_{\pm6.74}$ | $53.89_{\pm6.90}$ | $43.55_{\pm7.94}$ | $20.81_{\pm1.36}$ | $23.31_{\pm2.86}$ | $17.24_{\pm3.10}$ |
| GAT | $43.17_{\pm6.41}$ | $33.26_{\pm6.66}$ | $48.00_{\pm8.43}$ | $59.40_{\pm7.64}$ | $45.20_{\pm8.57}$ | $21.15_{\pm3.41}$ | $23.34_{\pm4.88}$ | $21.01_{\pm4.36}$ |
| GraphCL | $41.05_{\pm6.81}$ | $\underline{41.25}_{\pm8.24}$ | $47.21_{\pm7.80}$ | $52.37_{\pm7.28}$ | $42.80_{\pm8.15}$ | $21.40_{\pm3.21}$ | $25.48_{\pm3.97}$ | $20.87_{\pm4.68}$ |
| SimGRACE | $41.69_{\pm7.98}$ | $34.92_{\pm8.30}$ | $\underline{53.32}_{\pm8.42}$ | $62.39_{\pm7.83}$ | $52.05_{\pm9.43}$ | $21.63_{\pm3.16}$ | $22.85_{\pm4.11}$ | $19.16_{\pm5.20}$ |
| GPF+ | $38.22_{\pm6.97}$ | $38.53_{\pm7.80}$ | $49.34_{\pm6.22}$ | $53.88_{\pm7.60}$ | $45.42_{\pm9.28}$ | $21.85_{\pm3.78}$ | $26.09_{\pm5.03}$ | $20.47_{\pm4.58}$ |
| GraphPrompt | $38.44_{\pm7.39}$ | $37.75_{\pm8.61}$ | $52.57_{\pm7.84}$ | $54.57_{\pm7.98}$ | $42.21_{\pm7.62}$ | $21.66_{\pm3.36}$ | $25.94_{\pm4.54}$ | $20.67_{\pm4.64}$ |
| EdgePrompt+ | $40.66_{\pm7.66}$ | $32.19_{\pm6.23}$ | $44.72_{\pm6.17}$ | $60.67_{\pm8.35}$ | $50.90_{\pm9.34}$ | $20.78_{\pm1.87}$ | $25.00_{\pm3.98}$ | $20.53_{\pm4.31}$ |
| GCOPE | $36.85_{\pm6.62}$ | $34.79_{\pm8.08}$ | $52.69_{\pm7.52}$ | $59.18_{\pm9.10}$ | $53.28_{\pm9.60}$ | $\underline{22.34}_{\pm3.12}$ | $25.50_{\pm3.86}$ | $\underline{21.25}_{\pm3.96}$ |
| MDGPT | $\underline{43.38}_{\pm7.52}$ | $33.93_{\pm8.71}$ | $50.77_{\pm10.47}$ | $60.65_{\pm8.21}$ | $49.00_{\pm10.07}$ | $21.83_{\pm3.25}$ | $24.57_{\pm3.86}$ | $19.19_{\pm5.33}$ |
| SAMGPT | $40.33_{\pm8.32}$ | $36.79_{\pm8.72}$ | $49.38_{\pm5.95}$ | $63.77_{\pm9.50}$ | $51.14_{\pm8.65}$ | $20.83_{\pm2.14}$ | $24.72_{\pm3.97}$ | $17.88_{\pm4.69}$ |
| MDGFM | $41.45_{\pm8.27}$ | $37.58_{\pm8.99}$ | $51.73_{\pm7.47}$ | $\underline{67.35}_{\pm9.49}$ | $43.24_{\pm9.94}$ | $21.40_{\pm2.32}$ | $\underline{26.65}_{\pm4.86}$ | $17.06_{\pm5.21}$ |
| BRIDGE | $41.93_{\pm7.47}$ | $38.68_{\pm9.27}$ | $50.74_{\pm9.48}$ | $60.25_{\pm9.85}$ | $\underline{53.61}_{\pm9.86}$ | $21.40_{\pm3.21}$ | $25.67_{\pm4.22}$ | $21.66_{\pm4.79}$ |
| **MDGMIX** | $\mathbf{46.83}_{\pm7.77}$ | $\mathbf{44.56}_{\pm10.69}$ | $\mathbf{55.28}_{\pm7.73}$ | $\mathbf{68.40}_{\pm8.34}$ | $\mathbf{54.10}_{\pm10.22}$ | $\mathbf{22.92}_{\pm3.84}$ | $\mathbf{27.20}_{\pm4.75}$ | $\mathbf{23.32}_{\pm4.84}$ |

one domain; one is held out as the target domain while the remaining six are used for pre-training. For OGBN-Arxiv and Reddit, all seven datasets are used as source domains. Node features are aligned via PCA and projected to a 50-dimensional space, and a two-layer GCN is used as the encoder. In each training iteration, we construct 10 inter-domain and 10 intra-domain one-hop mixed subgraphs (hop=1), yielding 20 subgraphs in total.

For downstream evaluation, we conduct k-shot node and graph classification. We sample k labeled instances per class for training and use the rest for testing (Ren et al., 2018). For graph classification, node-centered subgraphs are extracted and inherit the label of the center node. Each k-shot setting is repeated 100 times with different label samplings. We report mean accuracy and standard deviation over these runs. For more detailed experiments and parameter settings, please refer to the Appendix A.2. For the code implementation, please refer to the supplementary materials.

### 6.2. RQ1: Performance on one-shot classification

We compare MDGMIX with all baseline models on one-shot node and graph classification tasks. As shown in Table 1, MDGMIX consistently outperforms all baselines under the one-shot setting. Methods based on pre-training and fine-tuning or graph prompting generally perform poorly in multi-domain scenarios, as they adapt only to the target domain and fail to leverage source-domain knowledge effectively. Compared to existing graph foundation model-based approaches, MDGMIX achieves consistent gains across all datasets. Notably, MDGMIX yields clear improvements on

challenging heterophilous graphs (Squirrel and Chameleon) as well as the large-scale OGBN-Arxiv, demonstrating robustness across diverse graph structures and scales. Unlike prior multi-domain graph methods that rely on learnable domain tokens, MDGMIX transfers only the pre-trained GNN and adapts it to the target domain via learnable weighting coefficients, resulting in improved memory and time efficiency during both pre-training and adaptation.

### 6.3. RQ2: Ablation Study

To evaluate the contribution of each component, we perform ablation studies during pre-training. Specifically, we construct MDGMIX variants by removing Boundary Mixup, Domain Classification, or Domain Decomposition, with Boundary Mixup replaced by random subgraph mixup when ablated. As shown in Table 3, removing any single component consistently degrades performance, confirming the necessity of all three modules. This also shows that the gains of MDGMIX do not simply come from subgraph mixup itself, but from selecting domain-ambiguous boundary regions for constructing more informative mixed samples. Moreover, the three components are complementary: Boundary Mixup provides challenging cross-domain samples, Domain Classification captures coarse inter-/intra-domain differences, and Domain Decomposition further separates shared and domain-specific signals. When both Boundary Mixup and Domain Classification are removed (Variant 4), performance drops but remains superior to existing baselines, indicating that Domain Decomposition alone can still enable effective cross-domain separation. In contrast, removing both Bound-

*Table 2.* Performance comparison on graph classification tasks (Accuracy). The best results are shown in bold and the runner-ups are underlined.

| Method | Cora | CiteSeer | Pubmed | Photo | Computers | Squirrel | Chameleon | OGBN-Arxiv |
|---|---|---|---|---|---|---|---|---|
| GCN | $41.36_{\pm11.42}$ | $30.67_{\pm8.62}$ | $50.09_{\pm9.63}$ | $56.26_{\pm9.77}$ | $47.91_{\pm11.62}$ | $20.70_{\pm1.49}$ | $22.64_{\pm3.63}$ | $17.36_{\pm5.18}$ |
| GAT | $43.39_{\pm10.79}$ | $33.57_{\pm9.52}$ | $50.78_{\pm9.30}$ | $55.12_{\pm9.99}$ | $48.07_{\pm11.10}$ | $20.52_{\pm1.75}$ | $22.78_{\pm3.93}$ | $17.03_{\pm4.88}$ |
| GraphCL | $46.81_{\pm8.13}$ | $40.38_{\pm7.21}$ | $51.65_{\pm8.11}$ | $58.12_{\pm8.61}$ | $49.99_{\pm10.97}$ | $20.32_{\pm2.57}$ | $26.45_{\pm4.52}$ | $20.72_{\pm3.98}$ |
| SimGRACE | $47.81_{\pm8.74}$ | $40.56_{\pm6.86}$ | $52.39_{\pm9.28}$ | $62.78_{\pm8.10}$ | $49.84_{\pm10.57}$ | $20.87_{\pm2.35}$ | $26.62_{\pm4.57}$ | $19.72_{\pm4.28}$ |
| GPF+ | $48.50_{\pm7.66}$ | $40.23_{\pm6.97}$ | $53.02_{\pm8.84}$ | $58.93_{\pm8.76}$ | $49.98_{\pm11.18}$ | $20.65_{\pm2.97}$ | $26.27_{\pm3.86}$ | $20.91_{\pm3.92}$ |
| GraphPrompt | $46.91_{\pm8.28}$ | $39.26_{\pm8.71}$ | $51.73_{\pm9.59}$ | $61.54_{\pm8.99}$ | $49.67_{\pm10.80}$ | $21.01_{\pm2.21}$ | $26.88_{\pm4.14}$ | $\underline{20.95}_{\pm4.20}$ |
| EdgePrompt+ | $42.87_{\pm8.47}$ | $37.67_{\pm7.20}$ | $48.49_{\pm8.64}$ | $62.55_{\pm8.68}$ | $50.76_{\pm11.46}$ | $21.04_{\pm2.69}$ | $27.00_{\pm4.87}$ | $20.37_{\pm3.89}$ |
| GCOPE | $45.69_{\pm9.30}$ | $40.03_{\pm7.24}$ | $53.67_{\pm10.40}$ | $58.42_{\pm8.09}$ | $49.05_{\pm11.96}$ | $\underline{21.21}_{\pm2.38}$ | $\underline{27.07}_{\pm5.06}$ | $18.19_{\pm4.27}$ |
| MDGPT | $47.24_{\pm8.52}$ | $39.06_{\pm10.11}$ | $50.77_{\pm10.29}$ | $58.91_{\pm10.05}$ | $47.79_{\pm10.81}$ | $21.10_{\pm2.05}$ | $25.73_{\pm5.26}$ | $17.72_{\pm4.18}$ |
| SAMGPT | $\underline{52.37}_{\pm9.16}$ | $41.15_{\pm8.88}$ | $50.42_{\pm10.59}$ | $62.06_{\pm8.79}$ | $47.53_{\pm9.93}$ | $20.84_{\pm2.23}$ | $26.67_{\pm4.70}$ | $20.54_{\pm3.79}$ |
| MDGFM | $50.87_{\pm9.39}$ | $\underline{46.51}_{\pm9.61}$ | $\underline{53.44}_{\pm11.63}$ | $\underline{64.53}_{\pm9.12}$ | $\underline{50.56}_{\pm11.55}$ | $21.06_{\pm2.05}$ | $26.79_{\pm4.65}$ | $18.72_{\pm4.16}$ |
| BRIDGE | $50.54_{\pm8.25}$ | $40.30_{\pm8.21}$ | $49.03_{\pm9.50}$ | $61.10_{\pm8.89}$ | $46.53_{\pm10.49}$ | $20.83_{\pm2.14}$ | $27.02_{\pm4.79}$ | $17.90_{\pm3.12}$ |
| **MDGMIX** | $\mathbf{52.81}_{\pm8.45}$ | $\mathbf{46.83}_{\pm9.54}$ | $\mathbf{56.42}_{\pm12.78}$ | $\mathbf{65.42}_{\pm8.22}$ | $\mathbf{52.23}_{\pm11.51}$ | $\mathbf{21.27}_{\pm2.12}$ | $\mathbf{27.46}_{\pm5.47}$ | $\mathbf{24.32}_{\pm3.78}$ |

*Table 3.* Ablation study of MDGMIX on node and graph classification tasks.

| Methods | Boundary Mixup | Domain Cls. | Domain Decomp. | Node Classification | | | Graph Classification | | |
|---|---|---|---|---|---|---|---|---|---|
| | | | | Cora | Citeseer | Photo | Cora | Citeseer | Photo |
| Variant 1 | × | ✓ | ✓ | $45.01_{\pm8.13}$ | $43.25_{\pm10.09}$ | $65.76_{\pm8.11}$ | $50.98_{\pm8.53}$ | $43.66_{\pm9.81}$ | $63.50_{\pm8.56}$ |
| Variant 2 | ✓ | × | ✓ | $44.22_{\pm8.28}$ | $43.13_{\pm9.29}$ | $64.62_{\pm7.88}$ | $51.40_{\pm9.17}$ | $45.87_{\pm8.75}$ | $64.90_{\pm8.45}$ |
| Variant 3 | ✓ | ✓ | × | $43.15_{\pm7.91}$ | $42.15_{\pm9.22}$ | $65.65_{\pm7.46}$ | $51.08_{\pm8.67}$ | $44.87_{\pm8.92}$ | $65.02_{\pm8.00}$ |
| Variant 4 | × | × | ✓ | $45.28_{\pm8.48}$ | $42.57_{\pm9.65}$ | $65.59_{\pm8.52}$ | $51.21_{\pm8.23}$ | $44.48_{\pm9.63}$ | $63.84_{\pm8.73}$ |
| Variant 5 | × | ✓ | × | $41.38_{\pm8.02}$ | $41.70_{\pm9.68}$ | $62.79_{\pm8.48}$ | $49.79_{\pm8.60}$ | $44.57_{\pm9.50}$ | $63.57_{\pm8.71}$ |
| **MDGMIX** | ✓ | ✓ | ✓ | $\mathbf{46.83}_{\pm7.77}$ | $\mathbf{44.56}_{\pm10.69}$ | $\mathbf{68.40}_{\pm8.34}$ | $\mathbf{52.81}_{\pm8.45}$ | $\mathbf{46.83}_{\pm9.54}$ | $\mathbf{65.40}_{\pm8.30}$ |

ary Mixup and Domain Decomposition (Variant 5) results in substantial performance degradation across all datasets, highlighting their critical role in disentangling transferable and domain-specific representations. Additional ablation results for the fine-tuning are reported in Appendix D.2.

## 6.4. RQ3: Efficiency Evaluation

To validate the efficiency of our method, we compare MDG-MIX with existing approaches in terms of pretraining time and memory consumption, as well as the number of trainable parameters during the adaptation stage. As shown in Figure 4a, MDGMIX consistently lies in the lower-left region of the memory–time space, achieving the best efficiency–capacity trade-off with the shortest pretraining time and memory footprint. Moreover, Figure 4b shows that MDGMIX introduces only a negligible number of fine-tuning parameters, whose scale is proportional to the number of source domains, indicating that efficient knowledge reuse can be achieved

with minimal adaptation cost. In contrast, existing methods typically rely on substantially increased computational budgets or extensive parameter updates. This efficiency mainly comes from the fact that MDGMIX pre-trains on a small set of boundary-centered mixed subgraphs rather than all source-domain graphs. Therefore, its computational cost depends more on the sampled subgraph budget than on the total size of the source domains. Such a design makes MDGMIX more suitable for scalable multi-domain graph pre-training when the number or size of source graphs increases. The time complexity analysis is in Appendix A.3.

## 6.5. RQ4: Hyperparameters Sensitivity

**Number of multi-domain mixing.** We also investigate multi-domain mixing by sampling domain weights from a Dirichlet distribution and mixing more than two domains. As reported in Table 4, increasing the number of mixed domains generally does not improve performance and may

*Table 4.* Performance of multi-domain mixing with different numbers of mixed domains.

| Mix Num. | Cora | Citeseer | Pubmed | Photo | Computers | Squirrel | Chameleon |
|----------|------|----------|--------|-------|-----------|----------|-----------|
| 3 | 46.20±7.90 | 43.40±9.62 | 52.76±9.43 | 64.67±8.29 | 52.68±10.91 | 22.80±3.65 | 27.17±4.84 |
| 4 | 45.41±7.57 | 43.52±9.68 | 55.79±9.42 | 64.29±8.41 | 51.07±10.41 | 22.59±3.82 | 26.32±4.54 |
| 5 | 45.06±7.58 | 43.24±9.90 | 52.34±8.73 | 66.67±7.91 | 50.84±11.36 | 22.86±3.76 | 26.97±4.48 |
| 6 | 45.47±8.11 | 41.66±9.74 | 53.80±8.50 | 65.77±7.66 | 49.60±10.23 | 23.04±4.52 | 27.44±4.91 |

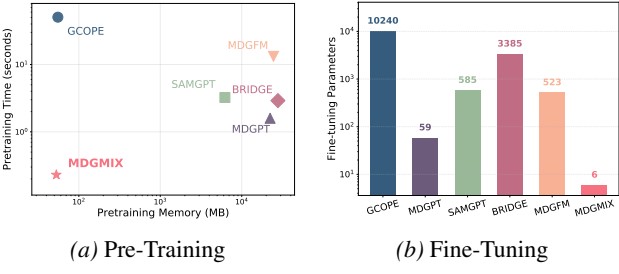

*(a)* Pre-Training      *(b)* Fine-Tuning

*Figure 4.* Analysis of pre-training and fine-tuning efficiency

*Table 5.* Effect of different Mixup coefficients. Larger $\alpha$ in Beta$(\alpha, \alpha)$ makes $\lambda$ more concentrated around 0.5. The fixed $\lambda = 0.5$ setting is stable and avoids tuning $\alpha$.

| Mixing Strategy | Cora | Computers | Chameleon |
|-----------------|------|-----------|-----------|
| $\lambda \sim \text{Beta}(0.2, 0.2)$ | 46.49±8.04 | 53.81±12.25 | 27.53±4.42 |
| $\lambda \sim \text{Beta}(1, 1)$ | 46.52±8.02 | 54.74±11.70 | 27.59±4.48 |
| $\lambda \sim \text{Beta}(10, 10)$ | 46.57±8.03 | 54.95±11.15 | 27.63±4.47 |
| Fixed $\lambda = 0.5$ | 46.55±8.05 | 54.95±11.12 | 27.62±4.46 |

even lead to degradation. This is likely because the fine-grained domain decomposition task becomes substantially harder as the number of mixed domains increases, since the model needs to recover a higher-dimensional and more entangled domain composition distribution. Therefore, MDG-MIX adopts pairwise cross-domain mixing, which is consistent with our boundary-pair construction strategy and provides a stable trade-off between sample diversity and decomposition reliability.

**Why fixed mixing coefficient $\lambda = 0.5$?** Although standard Mixup often samples $\lambda$ from a Beta distribution, MDG-MIX mixes subgraphs centered at boundary nodes that are already highly ambiguous across domains. In this case, a symmetric mixing coefficient prevents the mixed sample from collapsing toward one dominant domain and stabilizes the domain decomposition objective. Empirically, we compare $\lambda \sim \text{Beta}(\alpha, \alpha)$ with different $\alpha$ values. For a symmetric Beta distribution, a small $\alpha$ makes $\lambda$ concentrate near 0 and 1, producing samples close to one of the original domains, whereas a larger $\alpha$ makes $\lambda$ increasingly concentrated around 0.5, leading to more balanced cross-domain interpolation. We observe that performance becomes more stable as $\alpha$ increases and the sampled coefficients approach

0.5. Therefore, the fixed $\lambda = 0.5$ setting achieves comparable or slightly better performance while avoiding an additional hyperparameter, as shown in Table 5.

**Similarity threshold $\gamma$ and the boundary node ratio $\rho$.** We evaluated the sensitivity of MDGMIX to two critical hyperparameters: the cross-domain similarity threshold $\gamma$ and the boundary node ratio $\rho$. Figure 5 illustrates the performance surfaces for similarity threshold $\gamma$ and boundary ratio $\rho$ across three representative datasets. We observe that the accuracy surfaces are smooth and stable across a wide range of hyperparameters, indicating strong robustness. Notably, moderate values of $\gamma$ and $\rho$ consistently achieve near-optimal performance, suggesting that effective domain generalization stems from a balanced modeling of cross-domain similarity and boundary-aware representations. Detailed hyperparameter settings are provided in Appendix A.2.

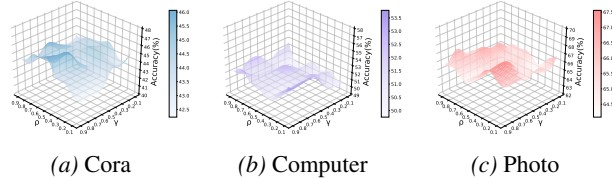

*(a)* Cora      *(b)* Computer      *(c)* Photo

*Figure 5.* Sensitivity of similarity threshold $\gamma$ and boundary rate $\rho$.

## 7. Conclusion

In this work, we proposed MDGMIX, an efficient multi-domain graph pre-training framework that addresses data redundancy in full-source-domain pre-training. By selecting domain-ambiguous boundary nodes and constructing compact mixed subgraphs, MDGMIX learns transferable cross-domain patterns with reduced computational cost. A hierarchical discrimination objective further helps decouple domain-invariant knowledge from domain-specific information, while the lightweight prompt-weighting mechanism enables efficient downstream adaptation with very few trainable parameters. Extensive experiments demonstrate that MDGMIX consistently improves few-shot node and graph classification performance while achieving better time and memory efficiency.

## Acknowledgements

This work was supported in part by the National Natural Science Foundation of China under Grants 62425605, 62133012, and 62303366, in part by the Key Research and Development Program of Shaanxi under Grants 2025CY-YBXM-041, 2022ZDLGY01-10, and 2024CY2-GJHX-15, and in part by the Fundamental Research Funds for the Central Universities and the Postgraduate Innovation Fund of Xidian University under Grant YJSJ26014.

## Impact Statement

This paper advances multi-domain graph representation learning by introducing MDGMIX, a lightweight pre-training framework for improving cross-domain generalization. By enabling efficient knowledge transfer across heterogeneous graph domains, MDGMIX supports scalable graph learning in applications such as social networks, recommendation systems, and biological networks. This work focuses on methodological advances in machine learning and does not raise specific ethical or societal concerns beyond those commonly associated with graph-based models.

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

# A. Experiment Details

## A.1. Dataset Statistics and Description

*Table 6.* Statistic of the multi-domain graph dataset.

| Dataset | Domain | Nodes | Edges | Feature Dimensions | Classes | Avg. Deg. |
|---|---|---|---|---|---|---|
| Cora | Academic | 2,708 | 10,556 | 1,433 | 7 | 3.90 |
| CiteSeer | Academic | 3,327 | 9,104 | 3,703 | 6 | 2.77 |
| PubMed | Academic | 19,717 | 88,648 | 500 | 3 | 4.50 |
| OGBN-Arxiv | Academic | 169,343 | 1,166,243 | 128 | 40 | 6.89 |
| Photo | E-Commerce | 7,650 | 238,162 | 745 | 8 | 31.13 |
| Computers | E-Commerce | 13,752 | 491,722 | 767 | 10 | 35.76 |
| Chameleon | Wikipedia | 2,277 | 36,101 | 2,325 | 5 | 31.71 |
| Squirrel | Wikipedia | 5,201 | 217,073 | 2,325 | 5 | 83.47 |
| Reddit | Social Network | 232,965 | 114,615,892 | 602 | 41 | 492.00 |

We evaluate MDGMIX on a diverse collection of graph benchmarks covering citation networks, e-commerce graphs, web graphs, and large-scale social networks. These datasets span different domains, graph sizes, feature dimensions, and structural properties. Detailed statistics of all datasets are summarized in Table 6.

**Citation Networks.** Cora, CiteSeer, and PubMed (Sen et al., 2008; Namata et al., 2012) are widely used citation network benchmarks. In these datasets, nodes represent scientific publications and edges denote citation relationships. Each node is associated with a bag-of-words or TF-IDF feature vector derived from the paper content, and the task is to predict the research topic of each paper. Specifically, Cora and CiteSeer contain publications from various computer science fields, while PubMed focuses on biomedical articles related to diabetes research. OGBN-Arxiv (Hu et al., 2020) is a large-scale citation graph from the Open Graph Benchmark (OGB). Nodes correspond to computer science papers indexed in arXiv, and edges represent citation links. Each node is described by a 128-dimensional feature vector extracted from paper titles and abstracts, and node labels indicate subject areas. Compared to classical citation datasets, OGBN-Arxiv is significantly larger and poses additional challenges in scalability and generalization.

**E-commerce Networks.** Photo and Computers (McAuley et al., 2015; Shchur et al., 2018) are Amazon co-purchase networks. In these graphs, nodes represent products and edges indicate that two products are frequently purchased together. Node features describe product metadata, and labels correspond to high-level product categories. These datasets capture user-driven relational patterns and exhibit relatively dense local connectivity.

**Wikipedia Networks.** Chameleon and Squirrel (Pei et al., 2020) are page-to-page networks derived from Wikipedia. Nodes represent web pages and edges correspond to hyperlinks between pages. Node features are constructed from textual content, typically using bag-of-words representations of nouns appearing on the pages. Labels are defined based on page categories or traffic patterns. These datasets are known to exhibit strong heterophily, where connected nodes often belong to different classes, making them particularly challenging for graph learning methods.

**Social Network.** Reddit (Hamilton et al., 2017) is a large-scale social network dataset. Nodes represent posts, and edges connect posts authored by the same user. Each node is associated with feature vectors derived from textual content, and labels correspond to discussion topics or communities. This dataset is characterized by its large size, high average node degree, and substantial class diversity.

## A.2. Experiment Setting

During pre-training, we adopt the Adam optimizer with both the learning rate and weight decay set to 0.0001. The hidden dimension of the GCN encoder is fixed to 256 for all methods. During downstream adaptation, the learning rate is set to 0.001. For all baseline methods, we use the optimal hyperparameters reported in their original papers. All experiments are

conducted on a single machine equipped with an NVIDIA 5090 GPU with 32GB memory, using PyTorch 2.8.0, CUDA 12.8, and PyG 2.6.1. Detailed hyperparameter settings for different datasets are summarized in Table 7.

*Table 7.* Hyperparameter settings for different datasets.

| Dataset | Pre-learning rate | Down-learning rate | Subgraph hop | Sample size | Boundary ratio $\rho$ | Similarity threshold $\gamma$ |
|---|---|---|---|---|---|---|
| Cora | 0.0001 | 0.001 | 1 | 10 | 0.1 | 0.7 |
| Citeseer | 0.0001 | 0.001 | 1 | 10 | 0.1 | 0.4 |
| Pubmed | 0.0001 | 0.001 | 1 | 10 | 0.3 | 0.5 |
| Computers | 0.0001 | 0.001 | 1 | 10 | 0.2 | 0.6 |
| Photo | 0.0001 | 0.001 | 1 | 10 | 0.3 | 0.5 |
| Chameleon | 0.0001 | 0.001 | 1 | 10 | 0.3 | 0.5 |
| Squirrel | 0.0001 | 0.001 | 1 | 10 | 0.4 | 0.8 |
| OGBN-Arxiv | 0.0001 | 0.001 | 1 | 10 | 0.4 | 0.3 |
| Reddit | 0.0001 | 0.001 | 1 | 10 | 0.3 | 0.6 |

### A.3. Time Complexity Analysis

Let $K$ denote the number of source domains and $d$ the unified feature dimension. For boundary node identification, we compute the distances between nodes and all domain centers, resulting in a complexity of $\mathcal{O}\left(\sum_k |V^{(k)}|Kd\right)$, with an additional $\mathcal{O}\left(\sum_k |V^{(k)}|K \log |V^{(k)}|\right)$ cost for top-$\rho$ boundary node selection. Since $K$ is typically small, this stage scales linearly with the total number of nodes. MDGMIX constructs only $N$ mixed subgraphs from a small set of boundary nodes. Extracting $h$-hop subgraphs incurs a cost of $\mathcal{O}(N(|V_h| + |E_h|))$, where $|V_h|$ and $|E_h|$ denote the average numbers of nodes and edges in the sampled subgraphs. The above operations are performed prior to pre-training, so they only need to be executed once.

Pretraining on $N_{intra}$ and $N_{inter}$ mixed subgraphs is dominated by GNN encoding. During cross-domain adaptation, the pretrained GNN is frozen, and only $K$ prompt weight parameters are optimized. Overall, MDGMIX avoids full-source-domain joint pretraining and scales with the number of sampled mixed subgraphs rather than the total size of all source-domain graphs, leading to substantially lower training cost compared with existing multi-domain graph foundation models.

## B. Additional Related Work

### B.1. Multi-Domain Graph Foundation Models

Multi-domain graph foundation models typically adopt a unified pre-training framework that learns general structural and semantic knowledge from multiple source-domain graphs, and subsequently transfer the model to unseen target domains to evaluate its generalization ability. This setting aims to integrate knowledge from diverse domains to support a wide range of downstream tasks across distribution shifts.

Existing methods generally follow a two-stage paradigm of multi-domain pre-training and cross-domain adaptation: the pre-training stage focuses on extracting cross-domain shared knowledge. In contrast, the adaptation stage aligns or adjusts the learned source-domain knowledge to the target domain. For example, GCOPE (Zhao et al., 2024) introduces virtual nodes to build bridges across domains, mitigating structural discrepancies among source graphs. MDGPT (Yu et al., 2024) assigns domain tokens to each domain to enhance feature discriminability and cross-domain alignability. SAMGPT (Yu et al., 2025) introduces structural tokens to align graph structural information across different domains. MDGFM (Wang et al., 2025) and SA2GFM (Shi et al., 2025) address the robustness of multi-domain graph learning. MDGFM enhances generalization to target domains by learning universal graph structures, while SA2GFM introduces an information bottleneck that leverages encoded tree text prompts and self-supervised information. BRIDGE (Yuan et al.) proposes Alignment Risk Regularization to strengthen the transferability of pre-trained models and adopts an MoE-based expert mechanism during adaptation to capture domain-specialized knowledge. GRAVER (Yuan et al., 2025) constructs a graph lexicon by extracting transferable subgraph patterns and achieves robust and efficient prompt fine-tuning through a generation-augmented lightweight routing

mechanism. RAG-GFM (Yuan et al., 2026) proposes a Retrieval-Augmented Generation framework that externalizes knowledge by constructing a dual-modal retrieval repository integrating semantic and structural information. By leveraging context augmentation, it achieves efficient downstream adaptation.

Despite the architectural innovations and sophisticated adaptation strategies proposed by these methods, they utilize the complete set of source-domain graph data during pre-training. This paradigm largely overlooks the substantial redundancy within multi-domain datasets, resulting in unnecessarily high pre-training costs and necessitating additional complex parameter modules in the adaptation stage to compensate for cross-domain discrepancies.

### B.2. Graph Mixup

Existing research has extended Mixup techniques (Zhang et al., 2017) to the graph domain and explored various mixing strategies. GraphMixup (Wang et al., 2021) introduces reinforcement mechanisms to dynamically adjust mixing ratios, mitigating class imbalance; GraphMix (Verma et al., 2021) blends node latent representations before feeding them into a fully connected layer; NodeMixup (Lu et al., 2024) designs inter-class and intra-class mixing separately for labeled and unlabeled nodes to alleviate underfitting; iGraphMix (Jeong et al., 2024) focuses on blending labeled node features and connections within the input graph; AGMixup (Lu et al., 2025) proposes a subgraph-based adaptive mixing strategy. IntraMix (Zheng et al., 2024) generates high-quality labels at low cost by blending and enhancing inaccurately labeled data within the same category, and enriches graph structures through high-confidence neighbor selection.

The aforementioned graph mixup technique primarily focuses on supervised scenarios involving single-domain graphs. We utilize domain categories as mixup labels and employ domain boundary-aware techniques to select domain-similar nodes for mixing.

## C. Theoretical Analysis and Proofs of MDGMIX

This section provides the complete theoretical analysis of MDGMIX, including all assumptions and detailed proofs. Our analysis is conducted entirely at the subgraph distribution level, which aligns with MDGMIX's boundary-aware subgraph mixing mechanism.

### C.1. Problem Setup and Notation

Let $\{P_k\}_{k=1}^K$ denote $K$ source-domain distributions over subgraphs $G \in \mathcal{G}_{\text{sub}}$, where each subgraph is defined as a fixed-radius ego-network extracted from a source graph. Each subgraph $G$ is associated with a label $Y \in \mathcal{Y}$. The target-domain distribution $P_t$ is not observed during training.

Let $f_\theta : \mathcal{G}_{\text{sub}} \to \mathcal{Z} \subset \mathbb{R}^d$ be a graph neural network (GNN) encoder, and let $g : \mathcal{Z} \to \mathcal{Y}$ be a task head. We define the hypothesis class $\mathcal{H} = \{h = g \circ f_\theta\}$.

For any distribution $P$ over subgraphs, the expected risk of $h \in \mathcal{H}$ is defined as:

$$\epsilon_P(h) = \mathbb{E}_{(G,Y) \sim P}[\ell(h(G), Y)]$$

where $\ell : \mathcal{Y} \times \mathcal{Y} \to [0, 1]$ is a bounded loss function.

### C.2. Core Assumptions

We now formally state all assumptions required for our theoretical analysis.

**Assumption C.1** (Lipschitz Encoder). The encoder $f_\theta$ is $L_f$-Lipschitz with respect to the subgraph edit distance $d_{\text{edit}}$, i.e.,

$$\|f_\theta(G_1) - f_\theta(G_2)\|_2 \le L_f \cdot d_{\text{edit}}(G_1, G_2), \quad \forall G_1, G_2 \in \mathcal{G}_{\text{sub}}.$$

Lipschitz Constant Estimation for GNNs: For an $L$-layer GCN with weight matrices $\{W^{(l)}\}_{l=1}^L$ and normalized adjacency matrix $\tilde{A}$, we have:

$$L_f \le \prod_{l=1}^L \sigma_{\max}(W^{(l)}) \cdot \|\tilde{A}\|_2^L$$

where $\sigma_{\max}(W^{(l)})$ denotes the largest singular value of $W^{(l)}$. Spectral normalization (Miyato et al., 2018) can be applied during training to enforce $L_f \leq 1$. Similar bounds hold for GIN and GraphSAGE encoders with appropriate normalization of the aggregation function.

**Assumption C.2** (Bounded and Lipschitz Loss). The loss function $\ell(\cdot, \cdot)$ satisfies:

1. Boundedness: $\ell \in [0, 1]$

2. Lipschitz continuity: $|\ell(z_1, y) - \ell(z_2, y)| \leq L_\ell \|z_1 - z_2\|_2, \quad \forall z_1, z_2 \in \mathcal{Z}, y \in \mathcal{Y}$

**Assumption C.3** (Structural Semantic Consistency). There exists a structural semantic space $\mathcal{S}$ and a mapping $\psi : \mathcal{Z} \to \mathcal{S}$ such that the push-forward distributions satisfy:

$$W_1(P_t^{\mathcal{S}}, R^{*\mathcal{S}}) \leq \epsilon_{\text{struct}}$$

where $P^{\mathcal{S}}$ denotes the distribution induced by $\psi$, and $\epsilon_{\text{struct}} \geq 0$ quantifies the structural distribution shift. This assumption relaxes label-space consistency and enables generalization to target domains with unseen or shifted label spaces.

**Assumption C.4** (Non-vanishing Boundary Mass). There exists $\rho_{\min} > 0$ such that for all domains $k$:

$$P_k(B_k) \geq \rho_{\min}$$

where $B_k$ denotes the boundary node set of domain $k$. This ensures sufficient boundary samples exist for effective mixing.

This assumption holds for most real-world graphs where nodes sufficiently far apart have negligible influence on each other's features and connections.

## C.3. Boundary-Aware Subgraph Mixing Operator

We now provide a mathematically rigorous definition of the boundary-aware subgraph mixing operator used in MDGMIX. Crucially, we strictly separate discrete topological operations from continuous feature interpolation.

**Definition C.5** (Boundary-Aware Subgraph Mixing Operator). Let $G_1 = (V_1, E_1, X_1)$ and $G_2 = (V_2, E_2, X_2)$ be two $h$-hop ego-subgraphs centered at boundary nodes from different domains. For $\lambda \in [0, 1]$, the mixed subgraph is defined as:

$$\mathcal{M}_\lambda(G_1, G_2) = (\ \underbrace{V_1 \cup V_2}_{\text{topology (discrete, }\lambda\text{-invariant)}}\ ,\ \underbrace{E_1 \cup E_2}_{\text{topology (discrete, }\lambda\text{-invariant)}}\ ,\ \underbrace{X_\lambda}_{\text{features (continuous, }\lambda\text{-dependent)}}\ ),$$

where the mixed feature matrix $X_\lambda \in \mathbb{R}^{|V_1 \cup V_2| \times d}$ is defined node-wise as:

$$X_\lambda(v) = \begin{cases} \lambda X_1(v) + (1 - \lambda) X_2(v), & \text{if } v \in V_1 \cap V_2 \quad \text{(overlap nodes)}, \\ X_1(v), & \text{if } v \in V_1 \setminus V_2 \quad \text{(boundary nodes from } G_1), \\ X_2(v), & \text{if } v \in V_2 \setminus V_1 \quad \text{(boundary nodes from } G_2). \end{cases}$$

**Remark.** The operator $\mathcal{M}_\lambda$ does not define a linear combination in the space of graphs. Instead, the mixing coefficient $\lambda$ induces a convex interpolation only in the node feature space, while the graph topology is merged discretely via set union. This design strictly adheres to the principle that continuous coefficients should not be improperly applied to discrete topological metrics (e.g., graph edit distance).

**Lemma C.6** (Representation Stability under Subgraph Mixing). *Let $f_\theta : \mathcal{G}_{\text{sub}}^{(h)} \to \mathbb{R}^d$ be an $L_f$-Lipschitz GNN encoder with respect to the subgraph edit distance $d_{\text{edit}}$ (Assumption C.1). For any boundary subgraphs $G_1, G_2$ and $\lambda \in [0, 1]$:*

$$\|f_\theta(\mathcal{M}_\lambda(G_1, G_2)) - f_\theta(G_1)\|_2 \ \leq \ (1 - \lambda) L_f \|X_2 - X_1\|_{V_1 \cap V_2} \ + \ L_f \cdot (|V_2 \setminus V_1| + |E_2 \setminus E_1|),$$

*where $\|X_2 - X_1\|_{V_1 \cap V_2} = \left(\sum_{v \in V_1 \cap V_2} \|X_2(v) - X_1(v)\|_2^2\right)^{1/2}$.*

*Proof.* We decompose the representation difference into two sequential operations:

**Step 1: Topology expansion.** Extend $G_1$ to the supergraph topology $(V_1 \cup V_2, E_1 \cup E_2)$ by adding nodes $V_2 \setminus V_1$ and edges $E_2 \setminus E_1$. Denote the resulting graph as $G_1^{\text{ext}} = (V_1 \cup V_2, E_1 \cup E_2, X_1^{\text{ext}})$, where

$$X_1^{\text{ext}}(v) = \begin{cases} X_1(v), & v \in V_1, \\ X_2(v), & v \in V_2 \setminus V_1. \end{cases}$$

The edit distance between $G_1$ and $G_1^{\text{ext}}$ equals the number of added nodes and edges:

$$d_{\text{edit}}(G_1, G_1^{\text{ext}}) = |V_2 \setminus V_1| + |E_2 \setminus E_1|.$$

By Assumption C.1:

$$\|f_\theta(G_1^{\text{ext}}) - f_\theta(G_1)\|_2 \leq L_f \cdot d_{\text{edit}}(G_1, G_1^{\text{ext}}) = L_f \cdot \left(|V_2 \setminus V_1| + |E_2 \setminus E_1|\right).$$

**Step 2: Feature interpolation.** On the fixed supergraph topology $(V_1 \cup V_2, E_1 \cup E_2)$, the feature difference between $G_1^{\text{ext}}$ and $\mathcal{M}_\lambda(G_1, G_2)$ is non-zero only on $V_1 \cap V_2$:

$$\|X_\lambda - X_1^{\text{ext}}\|_{V_1 \cap V_2} = (1 - \lambda) \cdot \|X_2 - X_1\|_{V_1 \cap V_2}.$$

Applying Lipschitz continuity again:

$$\|f_\theta(\mathcal{M}_\lambda(G_1, G_2)) - f_\theta(G_1^{\text{ext}})\|_2 \leq L_f \cdot (1 - \lambda) \cdot \|X_2 - X_1\|_{V_1 \cap V_2}.$$

**Step 3: Triangle inequality.** Combining Steps 1–2 via triangle inequality:

$$\begin{aligned} \|f_\theta(\mathcal{M}_\lambda(G_1, G_2)) - f_\theta(G_1)\|_2 &\leq \|f_\theta(\mathcal{M}_\lambda) - f_\theta(G_1^{\text{ext}})\|_2 + \|f_\theta(G_1^{\text{ext}}) - f_\theta(G_1)\|_2 \\ &\leq (1 - \lambda)L_f\|X_2 - X_1\|_{V_1 \cap V_2} + L_f\left(|V_2 \setminus V_1| + |E_2 \setminus E_1|\right). \end{aligned}$$

This completes the proof. $\square$

**Remark.** Unlike iGraphMix (Jeong et al., 2024) which uses random edge sampling to achieve "mixing" in topology space, MDGMIX employs deterministic set union for topology merging. Consequently, the mixing coefficient $\lambda$ affects only feature interpolation on overlapping nodes (scaled by $1 - \lambda$), while boundary node contributions form a $\lambda$-invariant constant term $L_f \cdot (|V_2 \setminus V_1| + |E_2 \setminus E_1|)$. This distinction is critical for theoretical soundness.

### C.4. Reference Distribution and Boundary Approximation

Following the reference-distribution framework for multi-source domain generalization, we define the ideal (infeasible) reference distribution:

$$R^* = \sum_{k,m=1}^{K} \alpha_{km} \text{Mix}_{\lambda=1/2}(P_k, P_m), \quad \sum_{k,m} \alpha_{km} = 1, \alpha_{km} \geq 0$$

where $\text{Mix}_\lambda(P_k, P_m)$ denotes the distribution of mixed subgraphs $\mathcal{M}_\lambda(G_k, G_m)$ with $G_k \sim P_k$ and $G_m \sim P_m$.

MDGMIX constructs a practical training distribution $\tilde{P}$ by mixing subgraphs centered strictly at boundary nodes.

**Definition C.7** (Mixed Subgraph Distribution). Let $\mathcal{B}^{(k)}$ denote the boundary node set of domain $k$. The mixed subgraph distribution $\tilde{P}$ is defined as:

$$\tilde{P} = \sum_{k,m=1}^{K} \alpha_{km} \text{Mix}_{\lambda=1/2}\left(P_k^{\mathcal{B}}, P_m^{\mathcal{B}}\right),$$

where $P_k^{\mathcal{B}}$ denotes the subgraph distribution restricted to boundary nodes $(\text{supp}(P_k^{\mathcal{B}}) = \{G_{v_i} \mid v_i \in \mathcal{B}^{(k)}\})$.

**Lemma C.8** (Boundary Approximation Error). *Define*

$$\Delta_{\text{bd}}^{(k)} = \sup_{G \in \text{supp}(P_k^{\text{int}})} \inf_{G' \in \text{supp}(P_k^{\text{bd}})} d_{\text{edit}}(G, G'),$$

where $P_k^{\text{int}}$ and $P_k^{\text{bd}}$ denote the interior and boundary subgraph distributions in domain $k$, respectively. Under Assumptions C.1 and C.4, the approximation error satisfies:

$$W_1(\tilde{P}, R^*) \ \leq \ C \cdot (1 - \rho_{\min}) L_f \, \mathbb{E}_k\Big[\Delta_{\text{bd}}^{(k)}\Big],$$

where $C > 0$ is an absolute constant independent of the domains.

*Proof.* By Assumption C.4, each source distribution admits the decomposition:

$$P_k = \rho_k P_k^{\text{bd}} + (1 - \rho_k) P_k^{\text{int}}, \quad \text{with } \rho_k \geq \rho_{\min}.$$

The reference distribution $R^*$ is a convex combination of mixture distributions constructed from $\{P_k\}_{k=1}^K$, and therefore contains mixture components supported on both boundary and interior subgraphs. In contrast, the practical distribution $\tilde{P}$ only retains mixture components supported on boundary subgraphs.

Consider transporting probability mass from $R^*$ to $\tilde{P}$. All mixture components involving interior-supported subgraphs must be mapped to boundary-supported ones. By definition of $\Delta_{\text{bd}}^{(k)}$, for any $G \sim P_k^{\text{int}}$ there exists $G' \sim P_k^{\text{bd}}$ such that

$$d_{\text{edit}}(G, G') \leq \Delta_{\text{bd}}^{(k)}.$$

Under Assumption C.1, the corresponding representation distance is bounded by $L_f \Delta_{\text{bd}}^{(k)}$.

Since the total probability mass associated with interior subgraphs is at most $1 - \rho_{\min}$, a feasible transport plan moves this mass to boundary-supported components with cost proportional to $L_f \Delta_{\text{bd}}^{(k)}$. Aggregating across all domains yields

$$W_1(\tilde{P}, R^*) \ \leq \ C \cdot (1 - \rho_{\min}) L_f \, \mathbb{E}_k\Big[\Delta_{\text{bd}}^{(k)}\Big],$$

where $C > 0$ is a constant that absorbs domain-pair mixing weights and transport couplings. This completes the proof. □

**Lemma C.9** (Lipschitz Risk Functional). *For any $h \in \mathcal{H}$, the mapping $\phi_h(G) = \ell(h(G), Y)$ is $L_\ell L_f$-Lipschitz with respect to $d_{edit}$.*

*Proof.* For any $G_1, G_2 \in \mathcal{G}_{\text{sub}}$:

$$\begin{aligned}
|\phi_h(G_1) - \phi_h(G_2)| &= |\ell(h(G_1), Y) - \ell(h(G_2), Y)| \\
&\leq L_\ell \|h(G_1) - h(G_2)\|_2 \quad \text{(by Assumption C.2)} \\
&= L_\ell \|g(f_\theta(G_1)) - g(f_\theta(G_2))\|_2 \\
&\leq L_\ell \|f_\theta(G_1) - f_\theta(G_2)\|_2 \quad \text{(assuming } g \text{ is 1-Lipschitz, standard for normalized heads)} \\
&\leq L_\ell L_f d_{\text{edit}}(G_1, G_2) \quad \text{(by Assumption C.1)}
\end{aligned}$$

Therefore, $\phi_h$ is $L_\ell L_f$-Lipschitz with respect to $d_{\text{edit}}$. □

**Lemma C.10** (Risk Difference Bound). *For any $h \in \mathcal{H}$:*

$$|\epsilon_{P_t}(h) - \epsilon_{\tilde{P}}(h)| \leq L_\ell L_f W_1(P_t, \tilde{P})$$

*Proof.* By the Kantorovich-Rubinstein duality theorem for Wasserstein distance:

$$W_1(P, Q) = \sup_{\|\phi\|_{\text{Lip}} \leq 1} |\mathbb{E}_{x \sim P}[\phi(x)] - \mathbb{E}_{x \sim Q}[\phi(x)]|$$

where $\|\phi\|_{\text{Lip}}$ is the Lipschitz constant of $\phi$.

Applying this to our risk functional with $\phi = \phi_h / \|\phi_h\|_{\text{Lip}}$ and using Lemma C.9:

$$\begin{aligned}
|\epsilon_{P_t}(h) - \epsilon_{\tilde{P}}(h)| &= |\mathbb{E}_{G \sim P_t}[\phi_h(G)] - \mathbb{E}_{G \sim \tilde{P}}[\phi_h(G)]| \\
&\leq \|\phi_h\|_{\text{Lip}} \cdot W_1(P_t, \tilde{P}) \\
&\leq L_\ell L_f W_1(P_t, \tilde{P})
\end{aligned}$$

completing the proof. □

## C.5. Concentration under Weak Dependence

Subgraphs extracted from the same graph are generally not independent due to node and edge overlap. Instead of assuming independence, we quantify statistical dependence through pairwise covariance.

**Definition C.11** (Subgraph Overlap Ratio). For two sampled subgraphs $G_i = (V_i, E_i)$ and $G_j = (V_j, E_j)$, we define their normalized node overlap ratio as

$$\text{ovlp}(G_i, G_j) = \frac{|V_i \cap V_j|}{\min\{|V_i|, |V_j|\}} \in [0, 1].$$

**Assumption C.12** (Overlap-controlled Dependence). There exists a constant $\kappa_{\text{ovlp}} \in [0, 1/4]$ such that for any $i \neq j$ and any $h \in \mathcal{H}$,

$$|\text{Cov}(Z_i, Z_j)| \leq \kappa_{\text{ovlp}} \, \text{ovlp}(G_i, G_j),$$

where $Z_i = \ell(h(G_i), Y_i) - \mathbb{E}[\ell(h(G_i), Y_i)]$.

**Definition C.13** (Dependence Measure). Let $Z_i = \ell(h(G_i), Y_i) - \mathbb{E}[\ell(h(G_i), Y_i)]$. Define:

$$\Delta_{\max} = \max_{i \neq j} \sup_{h \in \mathcal{H}} |\text{Cov}(Z_i, Z_j)| = \max_{i \neq j} \sup_{h \in \mathcal{H}} |\mathbb{E}[Z_i Z_j]|$$

**Corollary C.14** (Computable upper bound for $\Delta_{\max}$). *Under Assumption C.12 and with $\Delta_{\max}$ defined as in Definition C.13:*

$$\Delta_{\max} \leq \kappa_{\text{ovlp}} \cdot \max_{i \neq j} \text{ovlp}(G_i, G_j),$$

*which can be computed directly from the sampled subgraphs.*

*Proof.* By Definition C.13 and Assumption C.12:

$$\Delta_{\max} = \max_{i \neq j} |\text{Cov}(Z_i, Z_j)| \leq \max_{i \neq j} \kappa_{\text{ovlp}} \cdot \text{ovlp}(G_i, G_j) = \kappa_{\text{ovlp}} \cdot \max_{i \neq j} \text{ovlp}(G_i, G_j).$$

$\square$

**Lemma C.15** (Concentration with Weak Dependence). *For $n$ sampled mixed subgraphs $\{G_i\}_{i=1}^n$, with probability at least $1 - \delta$:*

$$|\epsilon_{\tilde{P}}(h) - \hat{\epsilon}_{\tilde{P}}(h)| \leq \sigma_{dep} \cdot \sqrt{\frac{\log(2/\delta)}{2n}}$$

*where $\sigma_{dep} = \sqrt{\frac{1}{4} + (n-1)\Delta_{\max}}$.*

*Proof.* Since the loss function $\ell$ is bounded in $[0, 1]$, the centered random variables $Z_i$ satisfy $|Z_i| \leq 1$ and $\text{Var}(Z_i) \leq \frac{1}{4}$. Let $S_n = \sum_{i=1}^n Z_i$. The variance of $S_n$ can be bounded as:

$$\text{Var}(S_n) = \sum_{i=1}^n \text{Var}(Z_i) + \sum_{i \neq j} \text{Cov}(Z_i, Z_j) \leq n \cdot \frac{1}{4} + n(n-1)\Delta_{\max} = n\left[\frac{1}{4} + (n-1)\Delta_{\max}\right]$$

Under this variance bound, a variance-controlled concentration inequality for weakly dependent bounded variables yields:

$$\Pr\left\{\left|\frac{1}{n}S_n\right| \geq t\right\} \leq 2\exp\left(-\frac{2nt^2}{\sigma_{\text{dep}}^2}\right)$$

where $\sigma_{\text{dep}}^2 = \frac{1}{4} + (n-1)\Delta_{\max}$ absorbs the pairwise dependence through $\Delta_{\max}$.

Setting the right-hand side equal to $\delta$ and solving for $t$:

$$t = \sigma_{\text{dep}}\sqrt{\frac{\log(2/\delta)}{2n}} = \sqrt{\frac{1}{4} + (n-1)\Delta_{\max}} \cdot \sqrt{\frac{\log(2/\delta)}{2n}}$$

Since $\frac{1}{n}\sum_{i=1}^n Z_i = \hat{\epsilon}_{\tilde{P}}(h) - \epsilon_{\tilde{P}}(h)$, we have with probability at least $1 - \delta$:

$$|\epsilon_{\tilde{P}}(h) - \hat{\epsilon}_{\tilde{P}}(h)| \leq \sigma_{\text{dep}} \cdot \sqrt{\frac{\log(2/\delta)}{2n}}$$

which completes the proof. $\square$

## C.6. Proof of Generalization Error Bound

**Theorem C.16** (Generalization Bound of MDGMIX). *For any $h \in \mathcal{H}$, with probability at least $1 - \delta$:*

$$\epsilon_{P_t}(h) \leq \hat{\epsilon}_{\tilde{P}}(h) + L_\ell L_f \cdot W_1(\tilde{P}, R^*) + L_\ell L_f \cdot \epsilon_{\text{struct}} + \epsilon_{\text{align}} + \sigma_{dep}\sqrt{\frac{\log(2/\delta)}{2n}}$$

*where $\sigma_{dep} = \sqrt{\frac{1}{4} + (n-1)\Delta_{\max}}$ and $\epsilon_{\text{align}}$ is defined as:*

$$\epsilon_{\text{align}} = \sup_{s \in \mathcal{S}} \left| \mathbb{E}_{Y \sim P_t(Y|s)}[\ell(h(z), Y)] - \mathbb{E}_{Y \sim R^*(Y|s)}[\ell(h(z), Y)] \right|.$$

*Proof.* Starting from Lemma C.10 and applying the triangle inequality:

$$\epsilon_{P_t}(h) \leq \epsilon_{\tilde{P}}(h) + L_\ell L_f W_1(P_t, \tilde{P})$$
$$\leq \epsilon_{\tilde{P}}(h) + L_\ell L_f\big(W_1(\tilde{P}, R^*) + W_1(R^*, P_t)\big)$$

By Lemma C.8, the boundary approximation error satisfies:

$$W_1(\tilde{P}, R^*) \leq C \cdot (1 - \rho_{\min})L_f \mathbb{E}_k[\Delta_{\text{bd}}^{(k)}]$$

For the transfer term between the reference distribution $R^*$ and the target distribution $P_t$, we proceed under Assumption C.3. There exists a structural semantic space $\mathcal{S}$ and a mapping $\psi : \mathcal{Z} \to \mathcal{S}$ such that the push-forward distributions satisfy $W_1(P_t^{\mathcal{S}}, R^{*\mathcal{S}}) \leq \epsilon_{\text{struct}}$.

Let $z = f_\theta(G)$ denote the subgraph representation produced by the GNN encoder, and $s = \psi(z)$ its corresponding structural semantic representation. We decompose the transfer gap as follows:

$$\epsilon_{P_t}(h) - \epsilon_{R^*}(h) = \mathbb{E}_{(G,Y) \sim P_t}[\ell(h(G), Y)] - \mathbb{E}_{(G,Y) \sim R^*}[\ell(h(G), Y)]$$
$$= \mathbb{E}_{s \sim P_t^{\mathcal{S}}}\big[\mathbb{E}_{Y \sim P_t(Y|s)}[\ell(h(z), Y)]\big] - \mathbb{E}_{s \sim R^{*\mathcal{S}}}\big[\mathbb{E}_{Y \sim R^*(Y|s)}[\ell(h(z), Y)]\big]$$
$$= \underbrace{\mathbb{E}_{s \sim P_t^{\mathcal{S}}}\big[\mathbb{E}_{Y \sim P_t(Y|s)}[\ell(h(z), Y)] - \mathbb{E}_{Y \sim R^*(Y|s)}[\ell(h(z), Y)]\big]}_{\epsilon_{\text{align}}}$$
$$+ \underbrace{\mathbb{E}_{s \sim P_t^{\mathcal{S}}}\big[\mathbb{E}_{Y \sim R^*(Y|s)}[\ell(h(z), Y)]\big] - \mathbb{E}_{s \sim R^{*\mathcal{S}}}\big[\mathbb{E}_{Y \sim R^*(Y|s)}[\ell(h(z), Y)]\big]}_{\text{(structural distribution shift)}}.$$

The first term $\epsilon_{\text{align}}$ captures the residual discrepancy between task semantics and structural semantics across domains. We do not impose additional assumptions on this term, as it reflects task-specific semantic mismatch that cannot be controlled by structural invariance alone.

For the second term, define $g(s) = \mathbb{E}_{Y \sim R^*(Y|s)}[\ell(h(z), Y)]$. We assume $\psi$ is non-expansive, so that $\|s_1 - s_2\| \leq \|z_1 - z_2\|$. Since $\ell$ is $L_\ell$-Lipschitz and $f_\theta$ is $L_f$-Lipschitz, the composition $z \mapsto \ell(h(z), Y)$ is $L_\ell L_f$-Lipschitz. Taking expectation over $Y|s$ preserves Lipschitz continuity as $\|\mathbb{E}[\phi_1] - \mathbb{E}[\phi_2]\| \leq \mathbb{E}[\|\phi_1 - \phi_2\|]$. Therefore, $g(s)$ is $L_\ell L_f$-Lipschitz with respect to $s$.

By the Kantorovich–Rubinstein duality of the Wasserstein distance, we obtain:

$$\mathbb{E}_{s \sim P_t^{\mathcal{S}}}[g(s)] - \mathbb{E}_{s \sim R^{*\mathcal{S}}}[g(s)] \leq L_\ell L_f \cdot W_1(P_t^{\mathcal{S}}, R^{*\mathcal{S}}) \leq L_\ell L_f \cdot \epsilon_{\text{struct}}.$$

Combining the above bounds yields:

$$\epsilon_{P_t}(h) - \epsilon_{R^*}(h) \leq \epsilon_{\text{align}} + L_\ell L_f \cdot \epsilon_{\text{struct}}.$$

Finally, by Lemma C.15, with probability at least $1 - \delta$:

$$|\epsilon_{\tilde{P}}(h) - \hat{\epsilon}_{\tilde{P}}(h)| \leq \sigma_{\text{dep}}\sqrt{\frac{\log(2/\delta)}{2n}}$$

Combining all these results:

$$\epsilon_{P_t}(h) \leq \epsilon_{\tilde{P}}(h) + L_\ell L_f \cdot W_1(\tilde{P}, R^*) + L_\ell L_f \cdot \epsilon_{\text{struct}} + \epsilon_{\text{align}}$$
$$\leq \hat{\epsilon}_{\tilde{P}}(h) + \sigma_{\text{dep}} \sqrt{\frac{\log(2/\delta)}{2n}} + L_\ell L_f \cdot W_1(\tilde{P}, R^*) + L_\ell L_f \cdot \epsilon_{\text{struct}} + \epsilon_{\text{align}}$$

This completes the proof. $\square$

*Remark* C.17 (Interpretation of Error Terms). The generalization bound decomposes the target error into five interpretable components:

- **Empirical risk** $\hat{\epsilon}_{\tilde{P}}(h)$: Training error on boundary-mixed subgraphs

- **Approximation error** $L_\ell L_f \cdot W_1(\tilde{P}, R^*)$: Gap between practical training distribution and ideal reference distribution, controlled by boundary sampling ratio $\rho_{\min}$

- **Structural shift** $L_\ell L_f \cdot \epsilon_{\text{struct}}$: Intrinsic discrepancy between source and target structural semantics (coefficient $L_\ell L_f$ is mandatory by KR duality)

- **Alignment error** $\epsilon_{\text{align}}$: Mismatch between structural roles and task labels across domains (dominant when label spaces differ)

- **Sampling variance** $\sigma_{\text{dep}} \sqrt{\log(2/\delta)/2n}$: Statistical error due to finite samples and weak dependence

This decomposition provides theoretical insight into why boundary-aware mixing can remain effective even when interior subgraphs exhibit high redundancy: the approximation error $W_1(\tilde{P}, R^*)$ is controlled by boundary mass $\rho_{\min}$ (Lemma C.8), suggesting that transferable knowledge concentrates near domain boundaries.

### C.7. Empirical Validation of Theoretical Assumptions

**Validation of structural–semantic consistency.** Assumption C.3 requires that boundary regions contain structurally and semantically transferable patterns across domains. To empirically examine this assumption, we conduct a domain classification experiment. Specifically, we train a GNN-based domain classifier on domain-center subgraphs, which are closest to the domain centroids and thus contain strong domain-specific biases. We then evaluate the classifier on three types of subgraphs: center subgraphs, random subgraphs, and boundary subgraphs.

As shown in Table 8, boundary subgraphs produce the lowest domain classification accuracy and the highest loss. Compared with center subgraphs, the relative accuracy drop on boundary subgraphs is approximately 32%. This indicates that boundary subgraphs are more difficult to assign to a specific source domain, supporting our claim that they lie in domain-ambiguous regions and are suitable for learning domain-invariant representations.

*Table 8.* Domain classification results on different node populations. Boundary subgraphs are the most domain-ambiguous.

| Subgraph Type | Domain Acc. | Loss |
|---|---|---|
| Center subgraphs | 83.33 | 0.4777 |
| Random subgraphs | 64.00 | 0.9012 |
| Boundary subgraphs | 56.67 | 1.1130 |

**Validation of non-vanishing boundary mass.** Assumption C.4 requires that each source domain contains a non-zero amount of boundary samples. We count the selected boundary nodes across domain pairs under strict boundary-selection settings. As shown in Table 9, all domain pairs contain non-zero boundary nodes. When the strict intersection of boundary candidates is empty in extreme cases, MDGMIX adopts the union-based fallback strategy described in Section 4.1, which selects the nearest cross-domain anchors and constructively ensures that boundary-centered subgraphs can still be generated.

*Table 9.* Number of selected boundary nodes across domain pairs under strict boundary-selection settings. Rows denote source domains and columns denote target domains.

| Source / Target | Cora | Citeseer | Pubmed | Computers | Photo | Squirrel | Chameleon |
|---|---|---|---|---|---|---|---|
| Cora | – | 25 | 25 | 25 | 25 | 25 | 26 |
| Citeseer | 31 | – | 32 | 32 | 32 | 32 | 32 |
| Pubmed | 190 | 193 | – | 192 | 192 | 193 | 191 |
| Computers | 128 | 128 | 128 | – | 122 | 122 | 122 |
| Photo | 72 | 73 | 69 | 66 | – | 68 | 67 |
| Squirrel | 47 | 49 | 49 | 48 | 47 | – | 48 |
| Chameleon | 18 | 16 | 17 | 17 | 17 | 16 | – |

## D. More Experiments

### D.1. Few-shot Classification.

We supplemented our node classification experiments with additional sample settings (3-shot, 5-shot, 10-shot), as shown in Table 10. MDGMIX achieved overall optimal performance across five benchmark datasets under all three settings—3-shot, 5-shot, and 10-shot—demonstrating stable and consistent advantages. Particularly in low-label scenarios (3-shot), MDGMIX achieves significant improvements over existing cross-domain pretraining methods (such as GCOPE, MDGPT, and MDGFM) on citation networks like Cora, CiteSeer, and PubMed. This demonstrates its ability to more effectively leverage cross-domain shared structural knowledge under minimal supervisory signals. From 3-shot to 10-shot settings, MDGMIX achieves an average accuracy improvement of 6.30% across five benchmarks, which is substantially larger than the strongest baseline MDGFM (3.94%). This indicates that MDGMIX benefits more effectively from additional supervision, suggesting that its boundary-aware subgraph mixing and hierarchical discrimination lead to more scalable and sample-efficient representations. We also provided the performance of different shots under graph classification, as shown in Table 11. Our approach also achieved competitive performance in graph classification.

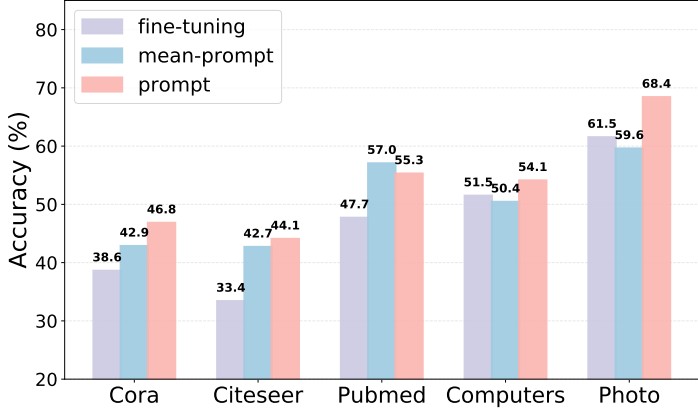

*Figure 6.* Ablation study of MDGMIX on adaptation phase

*Figure 7.* Large graph experiment on Reddit Dataset.

### D.2. Adaptation Ablation

We compare different prompting strategies, including no prompt, mean prompt, and learnable prompt. As shown in Figure 6, the pre-trained model achieves competitive performance even without prompts, indicating that the learned representations already exhibit strong generalization ability across domains. This observation suggests that the proposed multi-domain pre-training effectively captures transferable structural and semantic patterns, thereby reducing the dependence on task-specific adaptation. Mean prompting consistently improves performance over the no-prompt setting, despite introducing no additional learnable parameters. This result implies that a simple aggregation of domain-relevant information is sufficient

*Table 10.* Performance comparison on few-shot node classification tasks (Accuracy). The best results for each shot are shown in bold and the runner-ups are underlined.

| Method | Cora | CiteSeer | Pubmed | Photo | Computers |
|--------|------|----------|--------|-------|-----------|
| **3-shot** | | | | | |
| GCOPE | $53.78_{\pm 5.00}$ | $48.96_{\pm 4.70}$ | $\underline{61.92}_{\pm 5.20}$ | $76.58_{\pm 5.11}$ | $\underline{68.27}_{\pm 6.06}$ |
| MDGPT | $54.25_{\pm 5.46}$ | $48.88_{\pm 6.87}$ | $55.10_{\pm 4.99}$ | $69.73_{\pm 6.74}$ | $59.28_{\pm 8.14}$ |
| SAMGPT | $44.50_{\pm 6.30}$ | $53.08_{\pm 5.99}$ | $53.08_{\pm 5.94}$ | $74.06_{\pm 6.52}$ | $63.55_{\pm 7.88}$ |
| MDGFM | $\underline{57.61}_{\pm 4.77}$ | $\underline{57.05}_{\pm 6.01}$ | $60.13_{\pm 5.06}$ | $\underline{79.02}_{\pm 4.57}$ | $67.15_{\pm 5.98}$ |
| BRIDGE | $53.73_{\pm 5.02}$ | $55.66_{\pm 5.92}$ | $56.26_{\pm 5.73}$ | $65.91_{\pm 7.13}$ | $67.51_{\pm 7.61}$ |
| MDGMIX | $\mathbf{59.81}_{\pm 4.79}$ | $\mathbf{59.16}_{\pm 5.58}$ | $\mathbf{62.60}_{\pm 7.02}$ | $\mathbf{79.26}_{\pm 5.42}$ | $\mathbf{68.30}_{\pm 7.20}$ |
| **5-shot** | | | | | |
| GCOPE | $\underline{61.40}_{\pm 3.87}$ | $50.84_{\pm 4.85}$ | $\underline{63.62}_{\pm 5.29}$ | $79.44_{\pm 5.22}$ | $\underline{71.21}_{\pm 5.68}$ |
| MDGPT | $58.19_{\pm 4.73}$ | $54.42_{\pm 5.56}$ | $57.92_{\pm 3.95}$ | $73.79_{\pm 5.47}$ | $65.17_{\pm 7.70}$ |
| SAMGPT | $49.32_{\pm 6.04}$ | $57.81_{\pm 5.32}$ | $55.64_{\pm 5.40}$ | $77.32_{\pm 5.10}$ | $69.04_{\pm 5.26}$ |
| MDGFM | $60.25_{\pm 5.38}$ | $58.95_{\pm 5.72}$ | $62.32_{\pm 5.67}$ | $\underline{80.24}_{\pm 4.11}$ | $69.30_{\pm 5.95}$ |
| BRIDGE | $59.28_{\pm 4.63}$ | $\underline{60.56}_{\pm 4.22}$ | $58.61_{\pm 5.38}$ | $70.05_{\pm 6.06}$ | $71.12_{\pm 5.62}$ |
| MDGMIX | $\mathbf{62.69}_{\pm 3.97}$ | $\mathbf{62.39}_{\pm 3.95}$ | $\mathbf{64.58}_{\pm 5.75}$ | $\mathbf{81.14}_{\pm 3.91}$ | $\mathbf{72.10}_{\pm 4.44}$ |
| **10-shot** | | | | | |
| GCOPE | $65.82_{\pm 5.48}$ | $53.76_{\pm 5.92}$ | $\underline{66.03}_{\pm 5.82}$ | $\underline{81.99}_{\pm 5.74}$ | $73.69_{\pm 5.97}$ |
| MDGPT | $62.66_{\pm 3.94}$ | $61.36_{\pm 3.34}$ | $61.07_{\pm 4.27}$ | $74.69_{\pm 5.66}$ | $69.46_{\pm 4.49}$ |
| SAMGPT | $54.82_{\pm 4.72}$ | $63.15_{\pm 3.02}$ | $59.40_{\pm 5.68}$ | $77.32_{\pm 5.27}$ | $72.29_{\pm 4.20}$ |
| MDGFM | $62.87_{\pm 6.04}$ | $60.96_{\pm 5.63}$ | $64.27_{\pm 5.74}$ | $81.12_{\pm 3.83}$ | $71.42_{\pm 6.12}$ |
| BRIDGE | $63.56_{\pm 3.71}$ | $\underline{65.22}_{\pm 2.25}$ | $61.10_{\pm 6.01}$ | $70.71_{\pm 4.56}$ | $\underline{74.79}_{\pm 3.45}$ |
| MDGMIX | $\mathbf{68.26}_{\pm 3.18}$ | $\mathbf{65.45}_{\pm 2.17}$ | $\mathbf{67.36}_{\pm 2.92}$ | $\mathbf{84.01}_{\pm 3.03}$ | $\mathbf{75.57}_{\pm 2.41}$ |

to better align the pre-trained representations with downstream tasks. In contrast, learnable prompts further enhance performance by providing greater flexibility to adapt to domain shifts, albeit at the cost of additional parameters.

These results demonstrate that effective multi-domain pre-training can substantially reduce the reliance on complex and parameter-intensive adaptation mechanisms, while still allowing lightweight prompting strategies to yield consistent performance gains.

### D.3. Large Graph Experiment

We conducted experiments on large-scale social network datasets, as shown in Figure 7, comparing the performance of five multi-domain graph pre-training methods. The figure clearly demonstrates: MDGMIX achieved the best performance with an accuracy of 71.65%, significantly outperforming other methods. SAMGPT and BRIDGE followed with 69.87% and 69.01% accuracy, respectively, MDGPT and MDGFM demonstrated relatively lower performance at 67.81% and 67.62%, respectively.

### D.4. Hyperparameters Analysis

**Number of Hops.** Figure 8a examines the impact of neighborhood expansion depth by varying hop counts from 1 to 5. Model performance consistently peaks at 1-2 hops across diverse datasets before plateauing or declining. This unimodal pattern indicates that excessive neighborhood expansion introduces noise that dilutes boundary-specific structural patterns. These results validate our boundary-focused design principle while establishing practical guidelines for hop selection.

**Number of Samples.** Figure 8b demonstrates MDGMIX's remarkable data efficiency with respect to subgraph sampling quantity. Performance rapidly converges as sample size increases from 10 to 100, with diminishing returns beyond this threshold across all benchmark datasets. This observation directly corroborates our theoretical premise regarding substantial redundancy in multi-domain pretraining data. Even with merely 30 boundary-aware subgraphs, MDGMIX achieves 95% of

*Table 11.* Performance comparison on few-shot graph classification tasks (Micro-F1). The best results for each shot are shown in bold and the runner-ups are underlined.

| Method | Cora | CiteSeer | Pubmed | Photo | Computers |
|--------|------|----------|--------|-------|-----------|
| **3-shot** | | | | | |
| GCOPE | $63.58_{\pm5.10}$ | $49.88_{\pm6.18}$ | $\underline{63.12}_{\pm5.57}$ | $\underline{74.44}_{\pm4.12}$ | $62.68_{\pm7.44}$ |
| MDGPT | $54.53_{\pm6.16}$ | $48.17_{\pm6.95}$ | $57.80_{\pm7.60}$ | $66.58_{\pm7.69}$ | $56.33_{\pm8.30}$ |
| SAMGPT | $\underline{63.79}_{\pm4.89}$ | $52.73_{\pm5.88}$ | $60.91_{\pm6.21}$ | $72.56_{\pm6.07}$ | $60.48_{\pm7.73}$ |
| MDGFM | $63.45_{\pm6.00}$ | $\underline{58.59}_{\pm5.57}$ | $60.92_{\pm7.11}$ | $74.15_{\pm4.89}$ | $\underline{63.25}_{\pm7.08}$ |
| BRIDGE | $56.11_{\pm5.97}$ | $48.38_{\pm5.41}$ | $58.38_{\pm7.13}$ | $66.53_{\pm7.52}$ | $56.09_{\pm9.65}$ |
| MDGMIX | $\mathbf{65.09}_{\pm5.36}$ | $\mathbf{59.06}_{\pm5.65}$ | $\mathbf{64.13}_{\pm6.65}$ | $\mathbf{74.87}_{\pm5.65}$ | $\mathbf{63.40}_{\pm7.72}$ |
| **5-shot** | | | | | |
| GCOPE | $\underline{66.09}_{\pm4.99}$ | $52.80_{\pm6.03}$ | $65.49_{\pm5.48}$ | $\underline{76.16}_{\pm4.24}$ | $65.74_{\pm7.21}$ |
| MDGPT | $60.24_{\pm4.57}$ | $54.76_{\pm4.93}$ | $63.18_{\pm5.47}$ | $71.93_{\pm6.65}$ | $62.99_{\pm5.38}$ |
| SAMGPT | $65.31_{\pm2.74}$ | $58.77_{\pm3.96}$ | $\underline{66.07}_{\pm4.60}$ | $75.96_{\pm5.65}$ | $67.19_{\pm5.21}$ |
| MDGFM | $65.83_{\pm5.72}$ | $\underline{60.36}_{\pm5.01}$ | $63.11_{\pm6.50}$ | $75.52_{\pm4.35}$ | $\underline{67.70}_{\pm4.36}$ |
| BRIDGE | $59.26_{\pm4.83}$ | $55.42_{\pm4.79}$ | $62.74_{\pm5.62}$ | $71.35_{\pm6.95}$ | $63.35_{\pm7.03}$ |
| MDGMIX | $\mathbf{67.48}_{\pm4.98}$ | $\mathbf{61.47}_{\pm4.88}$ | $\mathbf{67.50}_{\pm5.53}$ | $\mathbf{76.56}_{\pm5.37}$ | $\mathbf{67.86}_{\pm7.49}$ |
| **10-shot** | | | | | |
| GCOPE | $\underline{68.93}_{\pm5.85}$ | $55.48_{\pm6.38}$ | $67.48_{\pm5.56}$ | $\underline{78.40}_{\pm4.44}$ | $68.47_{\pm7.20}$ |
| MDGPT | $65.46_{\pm3.93}$ | $61.49_{\pm3.14}$ | $67.03_{\pm3.54}$ | $72.27_{\pm6.32}$ | $66.90_{\pm3.83}$ |
| SAMGPT | $68.34_{\pm1.92}$ | $\mathbf{64.99}_{\pm1.97}$ | $68.77_{\pm3.13}$ | $78.25_{\pm5.28}$ | $69.98_{\pm2.89}$ |
| MDGFM | $67.33_{\pm2.29}$ | $62.09_{\pm4.89}$ | $\underline{69.50}_{\pm3.42}$ | $78.34_{\pm4.04}$ | $\underline{70.43}_{\pm3.46}$ |
| BRIDGE | $66.35_{\pm3.34}$ | $61.39_{\pm3.04}$ | $67.59_{\pm3.37}$ | $71.88_{\pm6.26}$ | $66.48_{\pm4.69}$ |
| MDGMIX | $\mathbf{69.67}_{\pm5.28}$ | $\underline{62.88}_{\pm4.57}$ | $\mathbf{69.76}_{\pm5.20}$ | $\mathbf{79.81}_{\pm5.75}$ | $\mathbf{70.98}_{\pm6.98}$ |

its peak performance on Cora and CiteSeer. These results demonstrate that effective multi-domain knowledge transfer does not necessitate exhaustive training on all available graph data.

### D.5. Comparison with Graph Mixup Baselines

Existing graph mixup methods are mainly designed for supervised or semi-supervised single-domain graph learning, where label information is required to construct meaningful mixed samples. In contrast, MDGMIX uses domain labels and boundary-aware cross-domain selection, which is more suitable for multi-domain pre-training under low-label downstream settings. We adapt representative graph mixup methods, including NodeMixup and iGraphMix, to the few-shot setting. As shown in Table 12, these methods do not perform well under extremely limited supervision, whereas MDGMIX consistently achieves the best results. This supports the necessity of boundary-aware domain-level mixing for multi-domain graph pre-training.

*Table 12.* Comparison with graph mixup methods on Computers under few-shot node classification.

| Method | 1-shot | 3-shot | 5-shot |
|--------|--------|--------|--------|
| GCN | $43.55\pm7.94$ | $61.87\pm4.33$ | $68.27\pm3.61$ |
| NodeMixup | $36.30\pm7.41$ | $61.63\pm6.17$ | $67.34\pm3.83$ |
| iGraphMix | $29.15\pm4.64$ | $45.98\pm8.84$ | $53.16\pm9.89$ |
| MDGMIX | $\mathbf{54.10\pm10.22}$ | $\mathbf{67.86\pm7.49}$ | $\mathbf{72.10\pm4.44}$ |

### D.6. Robustness to heterogeneous source domains.

We further construct highly divergent source-domain combinations involving heterophilic graphs such as Squirrel, Chameleon, Amazon-Ratings, and Roman-Empire, and evaluate transfer to homophilic target graphs. Despite large

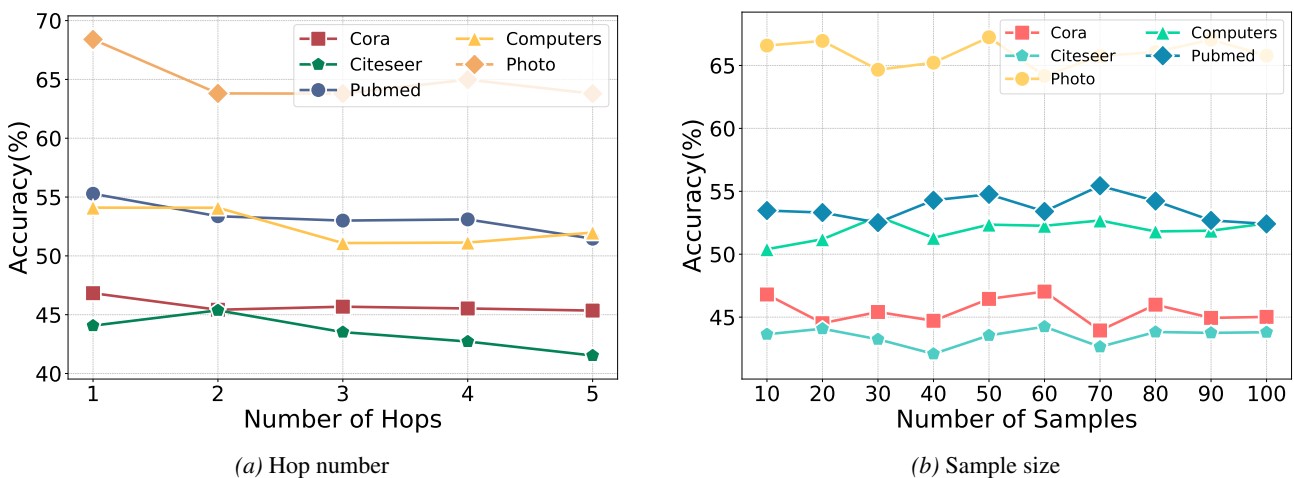

*(a)* Hop number          *(b)* Sample size

*Figure 8.* Hyperparameters analysis of the number of hops and sample sizes.

shifts in graph homophily and feature distributions, MDGMIX consistently outperforms competing methods, as shown in Table 13. This demonstrates that MDGMIX is robust to heterogeneous source-domain configurations.

*Table 13.* Robustness under highly heterogeneous source domains.

| Method | Cora | Citeseer | Computers |
|---|---|---|---|
| MDGPT | 35.49±6.60 | 30.38±8.09 | 47.15±8.11 |
| SAMGPT | 40.18±6.37 | 33.70±8.70 | 52.41±9.36 |
| MDGFM | 41.53±6.85 | 38.23±8.32 | 51.23±9.04 |
| MDGMIX | **42.80±8.52** | **44.16±9.97** | **53.30±8.93** |

### D.7. Domain Sensitivity

As shown in Table 14, by ablating different combinations of source domains, we observe that MDGMIX achieves the best and most stable performance across nearly all source-domain subsets on both Cora and Photo, reaching up to 45.17% on Cora and 66.17% on Photo, which demonstrates strong robustness to variations in source domains. In contrast, SAMGPT and MDGFM are more sensitive to the choice of source domains, exhibiting noticeable performance fluctuations; in particular, MDGFM suffers a significant degradation on the Photo target domain, with performance dropping as low as 55.20. Moreover, in some cases, removing certain source domains even leads to performance improvements, suggesting that multi-domain graph data may introduce conflicts or negative transfer rather than pure complementarity. By effectively mitigating inter-domain interference and extracting stable, transferable knowledge shared across domains, MDGMIX reduces reliance on any single source domain and maintains superior cross-domain generalization performance even under limited source availability.

### D.8. Visual Analytics

Figure 9 visualizes the learned node embeddings of different source domains and the target domain under three representative multi-domain pre-training methods. Circular markers denote nodes from source domains, while star-shaped markers correspond to target-domain nodes. We observe clear qualitative differences in the embedding geometry learned by different approaches.

For MDGPT (Figure 9a), embeddings from different source domains are heavily entangled, and target-domain nodes are scattered throughout the embedding space without forming coherent structures. This indicates that treating domains via explicit domain tokens is insufficient to align domain-invariant representations, leading to blurred inter-domain boundaries and unstable transfer to unseen domains.

SAMGPT (Figure 9b) partially improves domain separation by introducing structural alignment, as evidenced by the

*Table 14.* Performance of domain sensitivity analysis on different source domain datasets.

| Target domain | Source domain | | | | Method | | |
|---|---|---|---|---|---|---|---|
| | Citeseer | Pubmed | Squirrel | Computers | SAMGPT | MDGFM | MDGMIX |
| Cora | | | ✓ | ✓ | 38.81±5.79 | 41.03±9.23 | 41.97±9.09 |
| | ✓ | ✓ | | | 42.22±7.41 | 41.85±8.33 | 44.04±8.74 |
| | ✓ | ✓ | ✓ | | 41.63±6.60 | 40.65±8.03 | 44.69±9.01 |
| | ✓ | ✓ | ✓ | ✓ | 42.37±6.27 | 43.17±9.82 | 45.17±8.24 |
| Photo | | | ✓ | ✓ | 62.99±7.71 | 55.20±9.44 | 63.32±7.81 |
| | ✓ | ✓ | | | 64.23±8.01 | 55.78±7.78 | 66.17±9.33 |
| | ✓ | ✓ | ✓ | | 64.58±7.34 | 62.90±9.93 | 65.15±8.52 |
| | ✓ | ✓ | ✓ | ✓ | 63.58±8.11 | 60.34±8.46 | 63.52±8.89 |

*(a)* MDGPT  *(b)* SAMGPT  *(c)* MDGMIX

*Figure 9.* Embedding visualization of source and target domains, with circular nodes representing the source domain and star-shaped nodes representing the target domain.

emergence of several domain-dominated clusters. However, target-domain embeddings still concentrate in a limited region and remain weakly aligned with multiple source domains. This suggests that modeling domain differences alone may overemphasize domain-specific patterns, limiting effective cross-domain knowledge sharing.

In contrast, MDGMIX (Figure 9c) exhibits a more compact and structured embedding space. Source-domain nodes from different domains overlap substantially around shared regions, while still preserving mild domain-specific variations. Notably, target-domain nodes are well interleaved with source-domain clusters rather than isolated, indicating that MDGMIX successfully learns domain-invariant representations that generalize to unseen domains. This behavior is consistent with MDGMIX's boundary-aware subgraph mixing and hierarchical domain discrimination, which explicitly focus pre-training on domain-ambiguous regions and suppress spurious domain-specific prompts.

# E. Limitations

**Boundary Coverage and Domain Assumptions.** MDGMIX explicitly focuses pre-training on boundary-centered mixed subgraphs, motivated by both empirical redundancy observations and the theoretical analysis in Section 5. While this design significantly improves efficiency, it implicitly assumes that transferable knowledge is sufficiently concentrated near inter-domain boundaries. In scenarios where domain-specific information is distributed more uniformly or where boundary regions are sparse or poorly defined, restricting training to boundary nodes may omit useful structural patterns, potentially limiting performance gains.

**Sensitivity to Boundary Detection and Hyperparameters.** The effectiveness of MDGMIX depends on the quality of the selection of the boundary nodes, which is governed by hyperparameters such as the boundary ratio $\rho$ and the similarity threshold $\gamma$. Although our experiments show that MDGMIX is relatively stable within reasonable ranges, extreme settings may lead to either insufficient boundary coverage or overly noisy mixed subgraphs. Designing adaptive or data-driven

boundary selection strategies could further improve robustness across diverse domain configurations.

**Limited to Pairwise Cross-Domain Mixing.** The current instantiation of MDGMIX constructs inter-domain mixed subgraphs by combining boundary nodes from two source domains at a time. This pairwise design simplifies subgraph construction and stabilizes the hierarchical discrimination objective, but it does not explicitly model higher-order interactions among more than two domains. In settings where transferable knowledge emerges from the joint alignment of multiple domains simultaneously, restricting mixing to pairwise combinations may underexploit such multi-domain synergies. Extending MDGMIX to support multi-domain subgraph mixing is non-trivial, as it would require re-designing both the mixing mechanism and the fine-grained domain decomposition objective to handle higher-dimensional domain composition distributions, which we leave as future work.

