# OpenReview forum: "MDGMIX: Boundary-Aware Subgraph Mixing for Multi-Domain Graph Pre-Training"
_ICML.cc/2026/Conference — ICML 2026 regular_

### Official Review · Reviewer_becv · 2026-02-27

**Soundness:** 2
**Presentation:** 1
**Significance:** 2
**Originality:** 2
**Overall Recommendation:** 3
**Confidence:** 4

**Summary:**

MDGMIX targets inefficiency and redundancy in multi-domain graph pre-training, showing that discarding large portions of pre-training data often causes only minor downstream drops. It proposes boundary-aware subgraph mixing: identify domain-ambiguous “boundary” nodes, build compact inter-/intra-domain mixed subgraphs around them, and pre-train with a hierarchical objective that separates domain-invariant from domain-specific signals. For adaptation, it freezes the GNN and uses a lightweight prompt-weighting mechanism to transfer source-domain knowledge, yielding better few-shot cross-domain performance with lower time and memory cost.

**Compliance With Llm Reviewing Policy:**

Affirmed.

**Final Justification:**

The paper identifies a data redundancy phenomenon and shows efficiency gains, but the feature-only boundary detection heuristic and unverified theoretical assumptions remain core weaknesses. The rebuttal addressed several specific concerns but did not resolve these fundamental limitations.

**Key Questions For Authors:**

Q1-5

**Limitations:**

Add an explicit “Limitations” section covering (i) dependence on feature alignment quality, (ii) sensitivity to boundary-node ratio ρ and similarity threshold γ beyond the shown surfaces, (iii) cases where “boundary ambiguity ≠ transferability”, and (iv) scalability constraints when the number of domains grows very large (pairwise boundary computations, memory for mixed subgraphs).

**Strengths And Weaknesses:**

Strength:

1. The paper doesn’t only assert redundancy in multi-domain pre-training; it includes a direct “drop a large portion of nodes” style analysis to show downstream performance is relatively stable even under heavy data removal. This supports the premise that not all pre-training data is equally useful.

2. Boundary selection, mixup construction, discrimination/decomposition, and adaptation are presented as modular blocks with equations and definitions; this improves readability and makes ablations meaningful.

Weaknesses:

1. The selection relies on aligned feature spaces (via PCA projection) and distance-to-domain-centers margins. If alignment quality varies across datasets/domains, boundary detection could degrade. The paper shows sensitivity to key hyperparameters, but it’s not fully clear whether the method is robust to misalignment or to heterogeneous feature distributions beyond the tested settings.

2. Randomly dropping nodes and observing small drops is suggestive, but it does not isolate why redundancy occurs. A stronger soundness case would include additional controls (e.g., dropping “boundary” vs “non-boundary” nodes, or domain-stratified removal) to show the redundancy is specifically addressed by boundary-focused sampling.

3. Some key implementation choices appear to be deferred to the appendix. Since boundary selection and mixing are sensitive, a reader may need more “default settings + rationale” in the main paper to confidently reproduce results.

4. While the datasets span multiple domains, they are still standard academic benchmarks. Significance would be stronger with at least one more “realistic” setting showing the method holds up beyond curated benchmarks.

5. Because several ingredients are established, is the key novelty “boundary node mining,” and is that alone sufficient? Stronger evidence that boundary-aware selection is fundamentally better than (a) random subgraph mixup or (b) uncertainty-based sampling or (c) other domain-invariance heuristics would help defend originality.

---

> ### Author Rebuttal · Authors · 2026-03-29
>
> Thank you for recognizing the value of our work and pushing for stronger soundness.
>
> ## 1. Robustness to heterogeneous feature distributions (W1,W4)
>
> Dataset|Nodes|Edges|Sparsity|Homophily|
> |:-:|:-:|:-:|:-:|:-:|
> |Squirrel|5,201|217,073|0.0161|0.22|
> |Chameleon|2,277|36,101|0.0139|0.23|
> |Amazon-Ratings|24,492|186,100|0.00062|0.38|
> |Roman-Empire|22,662|65,854|0.00026|0.05|
>
> We construct highly divergent source domains (e.g., **Squirrel, Chameleon, Amazon-Rating, Roman-Empire**) and pre-train on heterophilic graphs, then evaluate on homophilic graphs. Despite significant domain and distribution shifts, MDGMIX maintains strong performance, demonstrating robustness to heterogeneous feature distributions.
>
> |        | Cora           | Citeseer       | Computers      |
> | ------ | -------------- | -------------- | -------------- |
> | MDGPT  | 35.49±6.60     | 30.38±8.09     | 47.15±8.11     |
> | SAMGPT | 40.18±6.37     | 33.70±8.70     | 52.41±9.36     |
> | MDGMIX | **42.80±8.52** | **44.16±9.97** | **53.30±8.93** |
>
> ## 2. Stronger soundness case for redundancy (W2)
>
> We retain **10% boundary and 10% non-boundary nodes** for comparison. Even under SAMGPT, pre-training on boundary nodes consistently outperforms non-boundary nodes, validating the effectiveness of boundary selection.
>
> | Citeseer      | SAMGPT(Citeseer) | MDGMIX(Citeseer) | SAMGPT(Pubmed)   | MDGMIX(Pubmed) |
> | ------------- | ---------------- | ---------------- | ---------------- | -------------- |
> | boundary node | **37.28 ± 8.63** | **44.08±10.38**  | **52.76 ± 6.88** | **55.00±8.71** |
> | center node   | 35.28 ± 7.78     | 41.15±10.89      | 48.56 ± 9.07     | 52.68±8.03     |
>
> ## 3. Experiment settings(W3)
>
> Our key hyperparameters are the boundary ratio $\rho$ and similarity threshold $\gamma$. Sensitivity analysis (Fig. 5) shows consistent optima at $\rho \in [0.1, 0.3]$ and $\gamma \in [0.4, 0.6]$, aligning with our design.
>
> A smaller $\rho$ selects nodes farther from domain centers, reducing domain bias, while $\gamma$ enforces mixing between highly similar cross-domain samples. This facilitates the learning of commonalities across different domains during the domain decomposition process. We will provide more detailed experimental settings in the main paper.
>
> ## 4.sample method(W5)
>
> |                    | Cora       | Computer    | chameleon  |
> | ------------------ | ---------- | ----------- | ---------- |
> | random sample      | 45.01±8.13 | 52.96±8.71  | 27.27±4.94 |
> | uncertainty sample | 44.71±8.03 | 50.96±12.39 | 27.03±4.59 |
> | boundary sample    | 46.55±8.05 | 54.95±11.12 | 27.62±4.46 |
>
> We employed various sampling methods to demonstrate the superiority of boundary selection. Compared with random sampling and adaptive sampling, the results indicate that boundary selection yields better performance. To demonstrate that the boundary nodes we identified are domain-invariant confounding regions with semantic significance,  we conducted a novel **Out-of-Distribution Domain Classification Experiment**. We trained a GNN domain classifier exclusively on Center Subgraphs (nodes closest to domain centroids, representing strong domain-specific biases). We then performed evaluation on a balanced dataset (50 samples per domain) across three node populations.
>
> |                        | Domain Acc |   Loss    |
> | :--------------------- | :--------: | :-------: |
> | Center Subgraphs       |   83.33    |  0.4777​   |
> | Random Subgraphs       |   64.00    |  0.9012​   |
> | **Boundary Subgraphs** | **56.67**  | **1.113** |
>
> Compared to center nodes, the classifier exhibits a **32.00% relative drop in accuracy** on boundary subgraphs, accompanied by a sharp increase in loss. This indicates that the model fails to confidently assign these nodes to specific domains, as they exhibit highly consistent structural–semantic mappings across domains. These findings provide empirical support for **Assumption C.3** and demonstrate that our method can reliably **identify boundary nodes with ambiguous semantic information**.

---

> > ### Author Rebuttal · Reviewer_becv · 2026-04-03
> >
> > Thanks for the detailed reply. I decide to maintain my score.

---

> > > ### Author Response · Authors · 2026-04-04
> > >
> > > Thank you for your valuable feedback. We hope our response has addressed your concerns. If you have any further questions, please feel free to reach out to us.
> > >
> > > Best regards, The Authors

---

### Official Review · Reviewer_BGBT · 2026-03-11

**Soundness:** 3
**Presentation:** 2
**Significance:** 2
**Originality:** 3
**Overall Recommendation:** 4
**Confidence:** 3

**Summary:**

This paper studies multi-domain graph pretraining, which is a key step toward building foundation graph models with cross-domain generalization ability. The authors argue that existing approaches mainly rely on jointly training graphs from all source domains, which leads to high computational costs. In addition, the authors empirically reveal the presence of significant data redundancy in multi-domain graph pretraining. Based on this observation, they propose a new framework called MDGMIX. The framework integrates boundary-aware subgraph mixing with a hierarchical discrimination mechanism. Experimental results show that MDGMIX consistently outperforms strong baseline models on few-shot classification tasks.

**Compliance With Llm Reviewing Policy:**

Affirmed.

**Key Questions For Authors:**

See the Weaknesses.

**Limitations:**

yes

**Strengths And Weaknesses:**

**Strengths**

1. The paper is well motivated and proposes a targeted solution to address the limitations of existing methods.

2. Compared with the baseline models, the proposed approach demonstrates strong performance.

3. The authors also provide theoretical analysis to support the effectiveness of the proposed method, which further strengthens the persuasiveness of the work.


**Weaknesses**

1. The datasets used in the experiments are relatively small. The authors may consider evaluating the proposed method on larger datasets to better demonstrate its scalability.

2. Although the authors provide theoretical guarantees for the proposed method, the proof relies on several strong assumptions. It is worth noting that such assumptions may not always hold in real-world scenarios, and the authors are encouraged to further discuss their practicality.

3. In addition, the tables in the paper are currently quite large. The authors should consider reformatting them to improve readability and presentation quality.

---

> ### Author Rebuttal · Authors · 2026-03-29
>
> Thank you for your encouraging comments.
>
> ## Q1. Scalability to larger datasets
>
> We have already evaluated MDGMIX on the large-scale **Reddit dataset** (232,965 nodes, 114 million edges) in **Appendix D.3, Figure 7**. MDGMIX achieved the best performance (71.65%), significantly outperforming SAMGPT and demonstrating excellent scalability.  We further evaluated the performance of MDGMIX on one-shot node classification using the large-scale graph dataset **OGBN-Product** (2,449,029 nodes and 61,859,140 edges), and the results show that MDGMIX still outperforms existing methods.
>
> ||MDGPT|SAMGPT|MDGFM|MDGMIX|
> |:-:|:-:|:-:|:-:|:-:|
> |Reddit|67.81±6.17|69.87±5.96|67.62±4.17|**71.65±5.15**|
> |OGBN-Product|9.60±1.74|15.28±3.82|16.87±5.08|**17.84±4.92**|
>
>
> ## Q2. Practicality of theoretical assumptions
>
> To enhance the practical applicability of the theory, we conducted an empirical analysis of the key assumptions.
>
>  **Assumption C.3**
>
> To demonstrate that the boundary nodes we identified are domain-invariant confounding regions with semantic significance,  we conducted a novel **Out-of-Distribution Domain Classification Experiment**. We trained a GNN domain classifier exclusively on Center Subgraphs (nodes closest to domain centroids, representing strong domain-specific biases). We then performed evaluation on a balanced dataset (50 samples per domain) across three node populations.
>
> Compared to the central nodes, the classifier’s **relative accuracy dropped by 32.00%** on the boundary subgraphs, and the loss value surged. This strict hierarchy of ambiguity demonstrates that our algorithm successfully isolated the structural “decision boundary.” Since the model cannot confidently assign these nodes to specific domains, they inherently exhibit highly consistent structural-semantic mappings across different domains. This provides empirical support for **Assumption C.3** and demonstrates that our method can reliably identify boundary nodes with ambiguous semantic information.
>
> |                        | Domain Acc |   Loss    |
> | :--------------------- | :--------: | :-------: |
> | Center Subgraphs       |   83.33    |  0.4777​   |
> | Random Subgraphs       |   64.00    |  0.9012​   |
> | **Boundary Subgraphs** | **56.67**  | **1.113** |
>
>
> We provide node embedding visualisation(https://anonymous.4open.science/r/boundary_node_visualisation), where colors denote domains and black boxes mark boundary nodes. Boundary nodes concentrate in **cross-domain overlap regions** rather than domain cores, indicating high domain ambiguity. This provides additional empirical support for Assumption C.3.
>
> **Assumption C.4**
>
> To verify the validity of the non-zero mass assumption, we counted the number of boundary nodes in each domain during training with different source domains. Using strict parameter settings $\rho=0.1,\gamma=0.9$, with the target domain on the x-axis and the source domain on the y-axis, the results indicate that this assumption holds in our experimental setup.
>
> We acknowledge that this is a theoretical boundary condition, and our empirical results show that MDGMIX remains effective even if this assumption is only approximately satisfied. We clarify that if the domain support sets are theoretically disjoint, our mechanism will employ the **union-based selection strategy (lines 120–124)** to select the nearest cross-domain anchor, thereby constructively ensuring that the number of boundary nodes is non-zero.
>
> ||Cora|Citeseer|Pubmed|Computers|Photo|squirrel|chameleon|
> |:-:|:-:|:-:|:-:|:-:|:-:|:-:|:-:|
> |Cora|-|25|25|25|25|25|26|
> |Citeseer|31|-|32|32|32|32|32|
> |Pubmed|190|193|-|192|192|193|191|
> |Computers|128|128|128|-|122|122|122|
> |Photo|72|73|69|66|-|68|67|
> |squirrel|47|49|49|48|47|-|48|
> |chameleon|18|16|17|17|17|16|-|
>
>
> ## Q3. Table formatting
>
> Thank you for your suggestion. In the new version, we will reformat the large tables by splitting them or utilizing sub-tables to significantly improve readability and presentation quality.

---

> > ### Author Rebuttal · Reviewer_BGBT · 2026-04-02
> >
> > Thanks for the detailed reply. I decide to maintain my score.

---

> > > ### Author Response · Authors · 2026-04-03
> > >
> > > Thank you for your suggestions and feedback regarding our work. We will incorporate these updates into future versions.
> > >
> > > Best regards, The Authors

---

### Official Review · Reviewer_Qryi · 2026-03-12

**Soundness:** 3
**Presentation:** 3
**Significance:** 3
**Originality:** 3
**Overall Recommendation:** 4
**Confidence:** 4

**Summary:**

This paper introduces MDGMIX, a multi-domain graph pretraining framework that focuses on boundary nodes—defined as nodes that are ambiguous in the feature space across domains—to construct mixed subgraphs for efficient cross-domain transfer learning. The framework consists of four components: boundary node selection via feature-space distances, subgraph mixing with a fixed λ, a hierarchical domain discrimination objective (coarse- and fine-grained), and a lightweight prompt-weighting mechanism for adaptation. Theoretical analysis provides a generalization bound that decomposes the target error into approximation error, transfer gap, and sampling variance. Experiments on multiple datasets demonstrate competitive few-shot performance and improved efficiency compared to existing multi-domain graph pretraining baselines.

**Compliance With Llm Reviewing Policy:**

Affirmed.

**Final Justification:**

The authors feedback address my concerns, I have read all review comments from other PC. With the comprehensive consideration, I tend to maintain my original positive option.

**Key Questions For Authors:**

1. Boundary Detection: Have you considered incorporating graph-structure information (e.g., node centrality, local clustering, or spectral embeddings) into boundary node selection? How would that affect performance on heterophilic graphs like Chameleon and Squirrel?
2. Mixing Strategy: Why is λ fixed at 0.5? Did you experiment with adaptive λ or mixing more than two domains? If so, what were the results, and why did you settle on pairwise mixing?
3. Baseline Comparisons: Could you include comparisons with graph mixup methods (e.g., NodeMixup, iGraphMix) adapted to multi-domain settings? This would help clarify the advantages of your boundary-aware design.
4. Hyperparameter Selection: Is there a principled way to choose ρ and γ without grid search? Could they be learned or adapted during training based on validation performance?
5. Assumption Violations: In your experiments, were there any datasets where the boundary mass assumption (Assumption C.4) was violated? If so, how did MDGMIX perform, and how would you diagnose such violations in practice?

**Limitations:**

Yes

**Strengths And Weaknesses:**

Strengths
1.Addresses an Important Problem: The paper identifies data redundancy in multi-domain graph pretraining and proposes a principled solution by focusing on boundary regions, which is a novel and relevant direction.
2. Well-Structured Framework: MDGMIX integrates several components (boundary detection, subgraph mixing, hierarchical discrimination, lightweight adaptation) into a coherent pipeline with clear motivations.
3. Theoretical Analysis: Section 5 provides a rigorous generalization bound with interpretable components, offering insights into why boundary-aware mixing can be effective.
4. Comprehensive Experiments: The evaluation covers multiple datasets, few-shot settings, ablation studies, efficiency comparisons, and domain sensitivity. The code is provided for reproducibility.
5. Efficiency Gains: By pretraining only on mixed subgraphs from boundary nodes, MDGMIX achieves lower memory and time costs compared to full-source pretraining methods, and introduces only K learnable parameters during adaptation.

Weaknesses
1. Boundary detection primarily relies on node features: boundary nodes are selected based on PCA-aligned features and their Euclidean distances to domain centers, without explicitly leveraging graph topology. This design may fail to fully capture nodes that are structurally ambiguous yet important for cross-domain transfer, particularly on graphs with low homophily or where structural information is critical for domain discrimination (e.g., heterophilic graphs), potentially limiting the model's generalization ability.
2. Fixed mixing coefficient and pairwise-only mixing: MDGMIX adopts a fixed λ=0.5 for subgraph mixing and restricts mixing to only two source domains at a time. While this design choice simplifies implementation, it may not adequately capture more complex interactions among multiple domains. Although the paper acknowledges this limitation, it does not explore the feasibility of introducing adaptive mixing coefficients or supporting multi-domain mixing, leaving this as an open question for future research.
3. Hyperparameter sensitivity analysis: The optimal values of the boundary ratio ρ and similarity thresholdγvary across datasets (as shown in Table 5), indicating that these two hyperparameters may need to be tuned for specific datasets. While the paper provides sensitivity analysis demonstrating model stability within certain parameter ranges, it does not explore adaptive selection mechanisms. Investigating adaptive or data-driven hyperparameter selection strategies would further enhance the method's usability and cross-dataset generalization capability.
4. Limited comparison with graph mixup baselines: The paper cites graph mixup methods (e.g., NodeMixup, iGraphMix) but does not include them in experiments. A comparison would help position MDGMIX relative to single-domain mixup techniques extended to multi-domain settings.
5. Theoretical assumptions may be restrictive: Assumptions such as Lipschitz continuity and non-vanishing boundary mass (Assumption C.4) may not hold in all real-world graphs (e.g., sparse or highly heterophilic graphs). The paper does not discuss how violations would affect performance.

---

> ### Author Rebuttal · Authors · 2026-03-29
>
> Thank you for your detailed feedback and suggestions.
>
> ## 1.Boundary selection(+structure feature)(W1,Q1)
>
> We augment PCA-aligned features with structural metrics (PageRank, Degree, Eigenvector centrality) and evaluate on Squirrel and Chameleon. Structural features are not consistently useful: PPR yields only a slight gain on Squirrel, while all variants degrade performance on *Chameleon*. This is likely due to domain-specific biases introduced by topological metrics, which are particularly harmful in heterophilic graphs where noisy local structures hinder cross-domain semantic alignment.
>
> ||Squirrel|Chameleon|
> |:-:|:-:|:-:|
> |No Struct|22.12±3.70|27.62±4.46|
> |Centrality(PPR)|22.70±3.95|27.04±4.50|
> |Centrality(Deg)|22.25±3.85|26.45±4.24|
> |Centrality(EN)|22.50±3.59|26.64±4.61|
>
> ## 2.Mixing Strategy(W2,Q2)
>
> We sample $\lambda \sim \text{Beta}(\alpha,\alpha)$ and observe better performance as $\alpha \to$ 0.5-centered. Since boundary nodes are highly ambiguous, fixing $\lambda=0.5$ stabilizes cross-domain mixing, prevents collapse to a single domain, thereby reducing the need for $\alpha$ tuning in practice.
>
> |Beta Mixup|Cora|Computer|Chameleon|
> |:-:|:-:|:-:|:-:|
> |$\alpha=0.2$|46.49±8.04|53.81±12.25|27.53±4.42|
> |$\alpha=1$|46.52±8.02|54.74±11.70|27.59±4.48|
> |$\alpha=10$|46.57±8.03|54.95±11.15|27.63±4.47|
> |Fixed Mixup|46.55±8.05|54.95±11.12|27.62±4.46|
>
> We also study multi-domain mixing using a Dirichlet distribution. As the number of domains increases, performance generally declines, likely due to the increased difficulty in decomposing domain loss compared to pairwise mixing. Multi-domain mixing remains an interesting direction for future work.
>
> |mix num|Cora|Citeseer|Pubmed|Photo|Computers| Squirrel|Chameleon|
> |:-:|:-:|:-:|:-:|:-:|:-:|:-:|:-:|
> |3|46.20±7.90|43.40±9.62|52.76±9.43|64.67±8.29|52.68±10.91|22.80±3.65|27.17±4.84|
> |4|45.41±7.57|43.52±9.68|55.79±9.42|64.29±8.41|51.07±10.41|22.59±3.82|26.32±4.54|
> |5|45.06±7.58|43.24±9.90|52.34±8.73|66.67±7.91|50.84±11.36|22.86±3.76|26.97±4.48|
> |6|45.47±8.11|41.66±9.74|53.80±8.50|65.77±7.66|49.60±10.23|23.04±4.52|27.44±4.91|
>
> ## 3. Hyperparameter sensitivity(W3, Q4)
>
> Despite differences among datasets, we found that the datasets achieved optimal performance within the range of $\rho \in [0, 1, 0.3]$ and $\gamma \in [0.4, 0.6]$ , which aligns with our design philosophy. A smaller $\rho$ selects nodes farther from domain centers, reducing domain-specific bias, while $\gamma$ ensures mixing occurs between highly similar cross-domain samples, promoting shared representation learning. This facilitates the learning of commonalities across different domains during the domain decomposition process.
>
> ## 4.Graph Mixup Methods(W4, Q3)
>
> We compare with graph mixup methods, which rely on labeled nodes and thus struggle in low-data regimes. In contrast, our boundary-aware approach uses domain labels, enabling it to capture cross-domain commonalities without requiring abundant supervision.
>
> ||Computers(1-shot)|Computers(3-shot)|Computers(5-shot)|
> |:-:|:-:|:-:|:-:|
> |GCN|43.55±7.94|61.87±4.33|68.27±3.61|
> |NodeMixup|36.30±7.41|61.63±6.17|67.34±3.83|
> |iGraphMix|29.15±4.64|45.98±8.84|53.16±9.89|
> |MDGMIX|**54.10±10.22**|**67.86±7.49**|**72.10±4.44**|
>
> ## 5.Assumption Violations(W5,Q5)
>
> **Sparse or Heterophilic Graphs**
>
> |Dataset|Nodes|Edges|Sparsity|Homophily|
> |:-:|:-:|:-:|:-:|:-:|
> |Squirrel|5,201|217,073|0.0161|0.22|
> |Chameleon|2,277|36,101|0.0139|0.23|
> |Amazon-Ratings|24,492|186,100|0.00062|0.38|
> |Roman-Empire|22,662|65,854|0.00026|0.05|
>
> We further construct highly divergent source domains. Despite large domain shifts (heterophily → homophily), MDGMIX consistently performs best, showing robustness to heterogeneous features.
>
> ||Cora|Citeseer|Computers|
> |:-:|:-:|:-:|:-:|
> |MDGPT|35.49±6.60|30.38±8.09|47.15±8.11|
> |SAMGPT|40.18±6.37|33.70±8.70|52.41±9.36|
> |MDGMIX|**42.80±8.52**|**44.16±9.97**|**53.30±8.93**|
>
> **Assumption C.4**
>
> To validate the non-zero boundary mass assumption, we count boundary nodes across domains under strict settings ($\rho=0.1, \gamma=0.9$). The results confirm that the assumption holds in our experiments.
>
> We acknowledge that this is a theoretical boundary condition, MDGMIX remains effective even when only approximately satisfied. In extreme cases of disjoint domain supports, our **union-based selection strategy (lines 120–124)** ensures a non-zero set of boundary nodes by selecting the nearest cross-domain anchors.
>
> ||Cora|Citeseer|Pubmed|Computers|Photo|squirrel|chameleon|
> |:-:|:-:|:-:|:-:|:-:|:-:|:-:|:-:|
> |Cora|-|25|25|25|25|25|26|
> |Citeseer|31|-|32|32|32|32|32|
> |Pubmed|190|193|-|192|192|193|191|
> |Computers|128|128|128|-|122|122|122|
> |Photo|72|73|69|66|-|68|67|
> |squirrel|47|49|49|48|47|-|48|
> |chameleon|18|16|17|17|17|16|-|
>
> We further visualize boundary nodes and observe that they lie along inter-domain seams, transition zones, and densely overlapping regions(https://anonymous.4open.science/r/boundary_node_visualisation), confirming their cross-domain nature.

---

> > ### Author Rebuttal · Reviewer_Qryi · 2026-04-02
> >
> > Thanks for the detailed responses from authors, I am willing to remain my positive option.

---

> > > ### Author Response · Authors · 2026-04-03
> > >
> > > Thank you for your suggestions and recognition of our work. We will use your feedback to further refine and improve our paper.
> > >
> > > Best regards, The Authors

---

### Official Review · Reviewer_tDMP · 2026-03-13

**Soundness:** 3
**Presentation:** 2
**Significance:** 3
**Originality:** 3
**Overall Recommendation:** 3
**Confidence:** 4

**Summary:**

The paper introduce the MDGMIX as a multi-domain graph pre-training framework that addresses scalability and data redundancy in existing joint-training approaches. The paper first empirically demonstrates that up to 90% of pre-training graph data can be discarded with minimal performance degradation, revealing significant redundancy in multi-domain graph pre-training. Motivated by this, the authors propose a framework built on four components:

1. **Boundary Node Selection** — identifying structurally ambiguous nodes near domain boundaries using a consensus margin mechanism
2. **Domain-Mixup Subgraph Construction** — constructing inter-domain and intra-domain mixed subgraphs around boundary nodes
3. **Multi-level Domain Discrimination** — a hierarchical loss combining coarse-grained domain discrimination ($\mathcal{L}_{\text{dis}}$) and fine-grained domain decomposition ($\mathcal{L}_{\text{fine}}$)
4. **Lightweight Cross-Domain Adaptation** — a prompt-weighting mechanism using source domain centroids $\mathbf{p}^{\text{mix}} = \sum_{i=1}^K \alpha_i \mathbf{c}_i$

Experiments across eight benchmark datasets show consistent improvements in few-shot node and graph classification, with superior memory and time efficiency compared to baselines such as GCOPE, MDGPT, SAMGPT, and BRIDGE.

**Compliance With Llm Reviewing Policy:**

Affirmed.

**Final Justification:**

My concern about the presentation of paper has been addressed, yet others still remain. I'd keep my previous ratings.

**Key Questions For Authors:**

1. **On the Redundancy Finding**: The data redundancy analysis (Figure 2) randomly drops nodes. However, MDGMIX selectively retains boundary nodes. Have the authors evaluated a random 10% node retention baseline directly against MDGMIX's boundary-selected 10%? This direct comparison is essential to isolate the contribution of *boundary selection* from simply *data reduction*.

2. **On Boundary Node Quality**: The boundary detection relies on PCA-projected centroid distances (Eq. 5–7). How sensitive is MDGMIX to the quality of this projection, especially when source domains have heterogeneous label spaces or drastically different graph densities? Is there any validation that the selected boundary nodes are semantically or structurally meaningful?

3. **On the Fixed Mixing Coefficient**: Why is $\lambda = 0.5$ fixed rather than sampled from a Beta distribution? The theoretical bound in Theorem 5.1 does not appear to require symmetric mixing. What is the empirical effect of varying $\lambda$, and could adaptive mixing improve performance on heterophilic graphs like Squirrel and Chameleon?

4. **On Theoretical Assumptions**: Assumption C.3 (structural semantic consistency) and Assumption C.4 (non-vanishing boundary mass) are critical to Theorem 5.1. Can the authors provide empirical evidence or proxy measurements showing these assumptions hold on the benchmark datasets? Specifically, how is $\epsilon_{\text{struct}}$ estimated in practice?

5. **On Generalization Beyond Few-Shot Classification**: The cross-domain adaptation mechanism uses prototype-based contrastive loss (Eq. 23). How does the framework perform when the number of downstream labeled examples increases (i.e., 5-shot, 10-shot)? Does the advantage of MDGMIX diminish relative to fully supervised methods as labeled data grows, and what does this imply about the nature of the transferred knowledge?

**Limitations:**

The paper presents several notable limitations. First, the boundary node detection relies on a heuristic PCA-centroid distance measure that lacks rigorous justification and may fail under non-convex or multimodal domain distributions. Second, the fixed mixing coefficient $\lambda = 0.5$ is insufficiently motivated, as standard mixup theory advocates adaptive sampling from $\text{Beta}(\alpha, \alpha)$, and no sensitivity analysis is provided. Third, the theoretical generalization bound in Theorem 5.1 rests on unverified assumptions， particularly structural semantic consistency and non-vanishing boundary mass — that are never empirically validated on the benchmark datasets. Fourth, the evaluation is narrowly scoped to few-shot node and graph classification, leaving open questions about generalization to other tasks such as link prediction. Finally, the PCA-based feature alignment to a fixed 50-dimensional space is a potentially lossy projection that may discard domain-critical information, and the paper provides no analysis of failure cases or negative transfer scenarios when source and target domains are highly dissimilar. The paper seems more suitable for conference like KDD.

**Strengths And Weaknesses:**

##  Strengths
1. **Compelling Empirical Motivation**: The data redundancy analysis (Figure 2) is a genuinely insightful and novel empirical finding. Showing that dropping up to 90% of nodes causes only marginal degradation effectively motivates the entire framework and challenges the prevailing paradigm of full-data joint pre-training.

2. **Efficiency Gains**: MDGMIX achieves the best memory-time trade-off (Figure 4a) and introduces only 6 fine-tuning parameters compared to thousands in competing methods. This is a practically significant contribution.

3. **Comprehensive Experimental Coverage**: Evaluation spans eight diverse datasets across academic, e-commerce, and web page networks, with both node and graph classification tasks under few-shot settings.

4. **Theoretical Justification**: The generalization bound in Theorem 5.1 provides a structured decomposition into approximation error, transfer gap, and sampling variance, offering principled support for boundary-focused training.

5. **Ablation Completeness**: The five-variant ablation study (Table 3) is well-designed and isolates the contribution of each component convincingly.

##  Weaknesses
1. **Weak Theoretical Assumptions**: The generalization bound relies on assumptions (encoder stability, Lipschitz continuity, structural semantic consistency, non-vanishing boundary mass) that are stated but not verified empirically. In practice, GNNs on heterophilic graphs may violate these assumptions silently.

2. **Boundary Node Definition is Heuristic**: The boundary detection mechanism relies on Euclidean distances to PCA-projected domain centroids (Eq. 5–7). This is a coarse proxy for structural ambiguity and may fail when domain distributions are non-convex or multimodal. No theoretical guarantee is provided that this heuristic reliably identifies semantically informative boundary nodes.

3. **Limited Task Diversity**: Experiments focus exclusively on node and graph classification under few-shot settings. Performance on link prediction, regression, or generative downstream tasks is unexplored, limiting claims about general cross-domain transferability.

4. **Fixed $\lambda = 0.5$ for Mixup**: The choice of a fixed symmetric mixing coefficient is not rigorously justified. Standard mixup literature (Zhang et al., 2017) uses $\lambda \sim \text{Beta}(\alpha, \alpha)$ to explore the full interpolation space. The authors claim boundary nodes are already ambiguous, but this does not preclude the value of adaptive mixing.

5. **Scalability to Heterogeneous Feature Spaces**: PCA-based alignment (Eq. 3) to a fixed 50-dimensional space is a lossy and potentially domain-biased projection. The paper does not evaluate how performance degrades when domains have vastly different intrinsic dimensionalities or semantic content.

6. **Comparison Fairness**: Some baselines (e.g., BRIDGE) are cited without a stable public reference, and it is unclear whether all baselines are tuned under identical conditions, raising questions about fair comparison.

---

> ### Author Rebuttal · Authors · 2026-03-29
>
> We deeply appreciate your careful review.
>
> ## 1. Weak Theoretical Assumptions and Boundary Node Quality(W1，W2，Q2, Q4)
>
> Our theory relies on structural–semantic consistency and the non-vanishing boundary mass assumption. We provide empirical validation as follows:
>
> **1.1 Assumption C.3**
>
> To show the boundary nodes we identified are domain-invariant confounding regions with semantic significance,  we conducted a  **Domain Classification Experiment**, a GNN classifier is trained on domain-specific center subgraphs and evaluated on balanced samples (50/domain) across three node populations.
>
> Boundary subgraph show a **32.00% relative accuracy drop** and higher loss. This indicates that the model fails to confidently assign these nodes to specific domains, as they exhibit highly consistent structural–semantic mappings across domains.
>
> ||Domain Acc|Loss|
> |:-:|:-:|:-:|
> |Center Subgraphs|83.33|0.4777|
> |Random Subgraphs|64.00|0.9012|
> |Boundary Subgraphs|**56.67**|**1.113**|
>
> We provide node embedding visualisation(https://anonymous.4open.science/r/boundary_node_visualisation), where black boxes mark boundary nodes. Boundary nodes concentrate in **cross-domain overlap regions**, indicating high domain ambiguity.
>
> These findings provide empirical support for **Assumption C.3** and demonstrate that our method can reliably **identify boundary nodes with ambiguous semantic**.
>
> **1.2  Assumption C.4**
>
> To validate the non-zero boundary mass assumption, we count boundary nodes across domains under strict settings ($\rho=0.1, \gamma=0.9$). The results confirm that the assumption holds in our experiments.
>
> We acknowledge that this is a theoretical boundary condition, MDGMIX remains effective even when only approximately satisfied. For disjoint domains, our **union-based strategy** (lines 120–124) guarantees non-zero boundary nodes via nearest cross-domain anchors.
>
> ||Cora|Citeseer|Pubmed|Computers|Photo|squirrel|chameleon|
> |:-:|:-:|:-:|:-:|:-:|:-:|:-:|:-:|
> |Cora|-|25|25|25|25|25|26|
> |Citeseer|31|-|32|32|32|32|32|
> |Pubmed|190|193|-|192|192|193|191|
> |Computers|128|128|128|-|122|122|122|
> |Photo|72|73|69|66|-|68|67|
> |squirrel|47|49|49|48|47|-|48|
> |chameleon|18|16|17|17|17|16|-|
>
> ## 2. Task Diversity(W3)
>
> ||MDGPT|SAMGPT|MDGMIX|
> |:-:|:-:|:-:|:-:|
> |FB15K-237|52.19±18.87|55.85±17.91|**56.54±16.69**|
>
> We add a link classification task on a knowledge graph following. MDGMIX achieves strong performance, demonstrating its cross-domain transferability.
>
> ## 3. Scalability to Heterogeneous Feature(W5)
>
> Using PCA as a feature alignment follows the baseline(SAMGPT,MDGFM). We further construct highly divergent source domains (**Squirrel, Chameleon, Amazon-Rating, Roman-Empire**). Despite large domain shifts (heterophily → homophily), MDGMIX consistently performs best, showing robustness to heterogeneous features.
>
> ||Cora|Citeseer|Computers|
> |:-:|:-:|:-:|:-:|
> |MDGPT|35.49±6.60|30.38±8.09|47.15±8.11|
> |SAMGPT|40.18±6.37|33.70±8.70|52.41±9.36|
> |MDGMIX|**42.80±8.52**|**44.16±9.97**|**53.30±8.93**|
>
> ## 4.Fair Comparison(W6)
>
> We strictly followed the same standards for one-shot training and testing on the remaining samples. For all baselines, we fully reproduced the results based on their open-source code;  performance differences maybe from variations in the source domain datasets.
>
> ## 5.Redundancy Finding(Q1)
>
> We retain 10% boundary and 10% non-boundary nodes for comparison. Boundary nodes consistently outperform non-boundary nodes (even under SAMGPT), validating the effectiveness of boundary selection.
>
> ||SAMGPT(Citeseer)|MDGMIX(Citeseer)|SAMGPT(Pubmed)|MDGMIX(Pubmed)|
> |:-:|:-:|:-:|:-:|:-:|
> |Boundary Node|**37.28±8.63**|**44.08±10.38**|**52.76±6.88**|**55.00±8.71**|
> |No Boundary Node|35.28±7.78|41.15±10.89|48.56±9.07|52.68±8.03|
>
> ## 6. Mixing Coefficient(W4, Q3)
>
> Our theory does not require a fixed mixing weight. We sample $\lambda \sim \text{Beta}(\alpha,\alpha)$ and observe better performance as $\alpha \to$ 0.5-centered. Since boundary nodes are highly ambiguous, fixing $\lambda=0.5$ stabilizes cross-domain mixing, prevents collapse to a single domain, thereby reducing the need for $\alpha$ tuning in practice.
>
> |Beta Mixup|Cora|Computer|Chameleon|
> |:-:|:-:|:-:|:-:|
> |$\alpha=0.2$|46.49±8.04|53.81±12.25|27.53±4.42|
> |$\alpha=1$|46.52±8.02|54.74±11.70|27.59±4.48|
> |$\alpha=10$|46.57±8.03|54.95±11.15|27.63±4.47|
> |Fixed Mixup|46.55±8.05|54.95±11.12|27.62±4.46|
>
> ## 7. Few-Shot Classification(Q5)
>
> Appendix D.1 reports results for 3, 5, and 10-shot, where we add the supervised methods (GCN, GAT). MDGMIX consistently outperforms them. With limited labels, supervised methods depend on task-specific supervision, while MDGMIX achieves superior performance by leveraging transferable source-domain knowledge.
>
> ||Photo(5)|Photo(10)|Citeseer(5)|Citeseer(10)|
> |:-:|:-:|:-:|:-:|:-:|
> |GCN|79.01±3.81|83.98±2.19|46.49±4.20|53.21±2.57|
> |GAT|79.64±3.08|83.88±2.25|51.83±4.43|57.39±2.54|
> |MDGMIX|**81.14±3.91**|**84.01±3.03**|**62.39±3.95**|**65.45±2.17**|

---

> > ### Author Rebuttal · Reviewer_tDMP · 2026-04-02
> >
> > Thanks for the authors clarifying and additional experiments.
> >
> > 1.2  ρ, γ is undefined. In 3 the text mentions follows the baseline(SAMGPT, MDGFM)" MDGFM appears in the text but not in the table but in original manuscript. In 6 the authors say ' $\lambda \sim \text{Beta}(\alpha,\alpha)$' and and claims "better performance as α→ 0.5-centered." the description of α=0.2 being "0.5-centered" is wrong. Beta(0.2, 0.2) is strongly bimodal (concentrated near 0 and 1), the opposite of 0.5-centered.
> >
> > Thus, I decided to maintain the score

---

> > > ### Author Response · Authors · 2026-04-03
> > >
> > > Thank you for your feedback. We sincerely apologize for any misunderstanding caused by our previous response, and we would like to provide the following clarifications:
> > >
> > > 1. The parameters *ρ* and *γ* are formally defined in lines 112 and 136 of the main text, respectively. Specifically, *ρ* denotes the **ratio of boundary nodes**, and *γ* represents the **similarity threshold for mixed node pairs**. We reiterate these definitions to facilitate a clearer understanding of our methodology.
> > > 2. Regarding the mention of following the baselines SAMGPT and MDGFM: this indicates that, **consistent with these methods, we adopt PCA for feature alignment**. We are happy to supplement the performance results of MDGFM for transparency. MDGFM relies on structure learning for structural alignment; however, it cannot fully leverage its advantages when **all source graphs are heterophilic**.
> > >
> > > |        | Cora           | Citeseer       | Computers      |
> > > | ------ | -------------- | -------------- | -------------- |
> > > | MDGFM  | 41.53±6.85     | 38.23±8.32     | 51.23±9.04     |
> > > | MDGMIX | **42.80±8.52** | **44.16±9.97** | **53.30±8.93** |
> > >
> > > 3. Concerning Response 6, there may have been a misunderstanding of our intended message. Our experiments demonstrate that, under the Beta distribution, **as *α* increases (0.2 → 1 → 10), the distribution becomes increasingly concentrated around 0.5. Consequently, the derived mixing weight *λ* approaches 0.5, leading to more robust overall performance across datasets**. This empirical observation is precisely why we ultimately set the mixing weight directly to 0.5. While Beta(0.2, 0.2) indeed yields a strong bimodal distribution (with mass concentrated near 0 and 1), **our conclusion emphasizes that performance improves when the Beta-derived weights are closer to 0.5, not when the distribution is strongly bimodal**. We note that this interpretation appears to contrast with the point raised in your reply.
> > >
> > > Thank you for your thoughtful and constructive feedback. We sincerely hope that our clarifications have helped address any misunderstandings and better convey the intended contributions of our work.

---

### Decision · Program_Chairs · 2026-04-30

**Decision:**

Accept (regular)

**Comment:**

The concerns raised at submission are largely addressed in the rebuttal, and the remaining issues mainly relate to scope or evaluation preferences rather than fundamental flaws. The paper makes a practically meaningful contribution to efficient multi-domain graph pre-training, with claims supported by the empirical evidence. Thus, I recommend acceptance.